# Epac2 in midbrain dopamine neurons contributes to cocaine reinforcement via enhancement of dopamine release

Xiaojie Liu[1], Casey R Vickstrom[1], Hao Yu[1], Shuai Liu[1], Shana Terai Snarrenberg[1], Vladislav Friedman[1], Lianwei Mu[1], Bixuan Chen[1], Thomas J Kelly[1], David A Baker[2], Qing-song Liu[1]*

[1]Department of Pharmacology and Toxicology, Medical College of Wisconsin, Milwaukee, United States; [2]Department of Biomedical Sciences, Marquette University, Milwaukee, United States

*For correspondence: qsliu@mcw.edu

**Abstract** Repeated exposure to drugs of abuse results in an upregulation of cAMP signaling in the mesolimbic dopamine system, a molecular adaptation thought to be critically involved in the development of drug dependence. Exchange protein directly activated by cAMP (Epac2) is a major cAMP effector abundantly expressed in the brain. However, it remains unknown whether Epac2 contributes to cocaine reinforcement. Here, we report that Epac2 in the mesolimbic dopamine system promotes cocaine reinforcement via enhancement of dopamine release. Conditional knockout of Epac2 from midbrain dopamine neurons (Epac2-cKO) and the selective Epac2 inhibitor ESI-05 decreased cocaine self-administration in mice under both fixed-ratio and progressive-ratio reinforcement schedules and across a broad range of cocaine doses. In addition, Epac2-cKO led to reduced evoked dopamine release, whereas Epac2 agonism robustly enhanced dopamine release in the nucleus accumbens in vitro. This mechanism is central to the behavioral effects of Epac2 disruption, as chemogenetic stimulation of ventral tegmental area (VTA) dopamine neurons via deschloroclozapine (DCZ)-induced activation of Gs-DREADD increased dopamine release and reversed the impairment of cocaine self-administration in Epac2-cKO mice. Conversely, chemogenetic inhibition of VTA dopamine neurons with Gi-DREADD reduced dopamine release and cocaine self-administration in wild-type mice. Epac2-mediated enhancement of dopamine release may therefore represent a novel and powerful mechanism that contributes to cocaine reinforcement.

## Editor's evaluation

This manuscript reports that Epac2, a downstream effector of cAMP, positively regulates cocaine reward by altering dopamine release properties in the striatum. The study uses a broad range of technical approaches to thoroughly characterize the cellular and behavioral roles of Epac2 in mice. Together, these results provide convincing important insight into Epac2 as a potential presynaptic molecular target through which dopamine signaling and drug taking might be manipulated and is of interest to scientists studying dopamine transmission and substance use disorders.

## Introduction

The rewarding effects of cocaine are primarily mediated by enhanced dopaminergic transmission in the mesolimbic dopamine system (*Anderson and Pierce, 2005*). Dopamine D$_1$- and D$_2$-like receptors are coupled to G$_{\alpha s}$ and G$_{\alpha i/o}$ proteins which lead to an increase or decrease in intracellular cAMP, respectively (*Tritsch and Sabatini, 2012*). Repeated cocaine exposure induces an upregulation of

cAMP signaling in the ventral tegmental area (VTA) and nucleus accumbens (NAc) (*Nestler, 2001*; *Self, 2004*; *Nestler, 2016*). In the VTA, repeated cocaine exposure has been shown to decrease $G_{\alpha i/o}$ expression (*Nestler et al., 1990*; *Striplin and Kalivas, 1992*), which results in elevated cAMP levels. Protein kinase A (PKA) and exchange protein directly activated by cAMP (Epac) are two major intracellular cAMP effectors (*de Rooij et al., 1998*; *Kawasaki et al., 1998*; *Tritsch and Sabatini, 2012*) and have numerous downstream signaling targets, thereby transducing changes in cAMP levels to cellular adaptations that can promote behavioral changes. The cAMP-PKA-CREB (cAMP response element-binding protein) signaling pathway has been extensively studied in the context of drug addiction (*Nestler, 2001*; *Self, 2004*; *Anderson and Pierce, 2005*; *Nestler, 2016*). However, few studies have addressed how the 'other' cAMP effector, Epac, regulates the cellular and behavioral effects of drugs of abuse.

There are two isoforms of Epac, Epac1 and Epac2. *Epac2* is predominately expressed in the brain, whereas *Epac1* expression in the brain is very low (*de Rooij et al., 1998*; *Kawasaki et al., 1998*). By making whole-cell recordings in ex vivo slices prepared from Epac1 and Epac2 global knockout mice (*Yang et al., 2012*), we have shown that Epac2, but not Epac1, regulates adaptations in excitatory and inhibitory synaptic plasticity in VTA dopamine neurons induced by non-contingent cocaine exposure (*Liu et al., 2016*; *Tong et al., 2017*). Drug self-administration provides a measure of the reinforcing efficacy of drugs of abuse and an animal's motivation to acquire rewarding drugs (*O'Connor et al., 2011*; *Hiranita, 2015*). The overall goal of this study was to determine whether Epac2 signaling in midbrain dopamine neurons regulates cocaine self-administration. Epac activators enhance presynaptic glutamate release by increasing the association between the active zone proteins Rab3A, RIM and Munc13, thereby facilitating vesicle docking and exocytosis (*Ozaki et al., 2000*; *Ferrero et al., 2013*; *Fernandes et al., 2015*). Axonal dopamine release in the striatum is mediated by vesicular exocytosis and requires active zone-like site machinery (*Liu and Kaeser, 2019*). Dopamine neuron-specific knockout of RIM abolished action potential-triggered dopamine release in the striatum (*Liu et al., 2018a*). However, no previous studies have examined whether Epac2 modulates dopamine release.

In the present study, we generated dopamine neuron-specific Epac2 conditional knockout (Epac2-cKO) mice and employed the Epac2-selective antagonist ESI-05 (*Tsalkova et al., 2012*) to investigate the role of Epac2 signaling in midbrain dopamine neurons in regulating cocaine reinforcement. We examined the effects of ESI-05 and Epac2-cKO on fixed ratio-1 (FR1) and progressive ratio (PR) reinforcement schedules and cocaine dose-response curves. As the reinforcing efficacy of cocaine is primarily mediated by its effect to enhance extracellular dopamine levels (*Anderson and Pierce, 2005*), we tested the hypothesis that Epac2 may contribute to cocaine reinforcement by modulating dopamine release. We first determined how pharmacological and genetic manipulations of Epac2 affect dopamine release in the striatum. Next, we expressed Gs- and Gi-designer receptors exclusively activated by designer drugs (DREADDs) (*Roth, 2016*) in VTA dopamine neurons and examined whether bidirectional modulation of dopamine release with DREADD activation affected cocaine self-administration in wild-type (WT) and Epac2-cKO mice. Together, our studies provide evidence for a critical role of Epac2 in regulating dopamine release and cocaine reinforcement.

## Results

### Epac2 inhibition decreased cocaine self-administration across multiple reinforcement schedules and cocaine doses

We examined the effect of the selective Epac2 inhibitor ESI-05 on cocaine self-administration in roughly equal numbers of male and female mice using an Instech self-administration system that greatly extends the longevity of catheter patency (*Slosky et al., 2020*; *Slosky et al., 2021*). C57BL/6J mice underwent training for cocaine self-administration under a FR1 reinforcement schedule (0.5 mg/kg/infusion) in 3 hr sessions over 10 days, and 18 of 20 mice with patent catheters acquired stable cocaine self-administration (criteria are detailed in Materials and methods). The average number of sessions to reach stable acquisition was 5.4±0.3 sessions (*Figure 1A*). On subsequent days, mice were injected with ESI-05 (3.3 or 10 mg/kg, i.p.) or vehicle (0 mg/kg, i.p.) 10 min prior to self-administration testing in a Latin Square design. ESI-05 dose-dependently reduced active nose pokes (*Figure 1B*) and cocaine infusions (*Figure 1C*) but had no significant effect on inactive nose pokes (*Figure 1D*; detailed

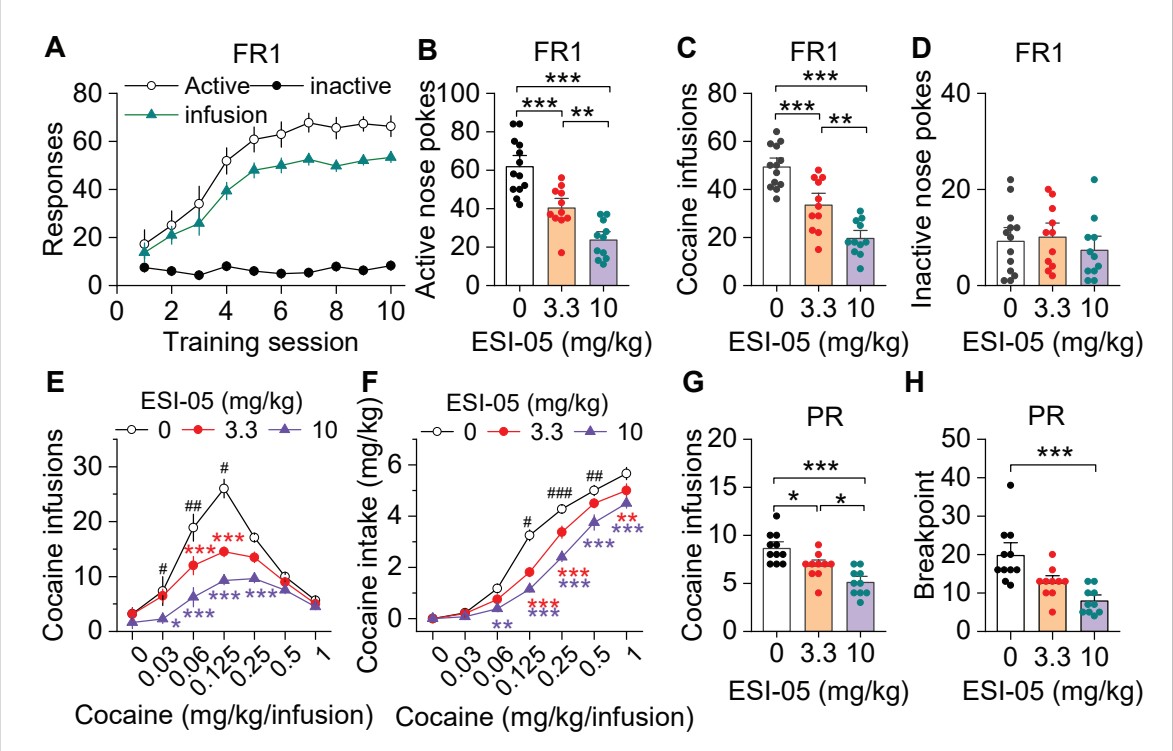

**Figure 1.** ESI-05 dose-dependently decreased cocaine self-administration under fixed ratio-1 (FR1) and progressive ratio (PR) reinforcement schedules in mice. (**A**) Active nose pokes and inactive nose pokes and cocaine infusions during 10-day FR1 self-administration training (0.5 mg/kg/infusion, n=18, 9 male and 9 female). The FR1 schedule has a brief timeout (10 s) period following each active nose poke during which additional nose pokes do not result in cocaine infusions. As a result, the number of nose pokes is higher than the number of cocaine infusions. (**B–D**) The mean number of active nose pokes (one-way ANOVA: $F_{2,32}$ = 30.9, p<0.001), cocaine infusions (one-way ANOVA: $F_{2,32}$=31.9, p<0.001), and inactive nose pokes (one-way ANOVA: $F_{2,32}$ = 0.5, p=0.613) in response to ESI-05 pretreatments (0, 3.3, 10 mg/kg, i.p.) under an FR1 reinforcement schedule (n=11–13, 6 male and 5–7 female per group). (**E–F**) ESI-05 pretreatments produced a significant downward shift in the dose-response curve for cocaine infusions (two-way repeated-measures ANOVA: ESI-05 treatment, $F_{2,22}$ = 45.5, p<0.001; cocaine dose, $F_{6,132}$ = 51.0, p<0.001; ESI-05 treatment × cocaine dose interaction, $F_{12,132}$ = 6.5, p<0.001; 3.3 vs. 0 mg/kg ESI-05, red *; 10 vs. 0 mg/kg ESI-05, violet *; 3.3 vs. 10 mg/kg ESI-05 black #) and decreased total cocaine intake (two-way repeated-measures ANOVA: ESI-05 treatment, $F_{2,22}$ = 66.5, p<0.001; cocaine dose, $F_{6,132}$ = 498.1, p<0.001; ESI-05 × cocaine dose interaction, $F_{12,132}$ = 6.7, p<0.001; 3.3 vs. 0 mg/kg ESI-05, red *; 10 vs. 0 mg/kg ESI-05 violet *; 3.3 vs. 10 mg/kg ESI-05 black #) on the dose-response curve (n=8–9, 5 male and 4 female). (**G–H**) ESI-05 dose-dependently decreased the number of cocaine infusions (one-way ANOVA: $F_{2,28}$ = 16.1, p<0.001; n=10–11, 5 male and 5–6 female) and breakpoint (Brown-Forsythe equal variance: p<0.05; Kruskal-Wallis one-way ANOVA on ranks: ESI-05, H=17.5, p<0.001; n=10–11, 5 male and 5–6 female) under a PR reinforcement schedule. *p<0.05, **p<0.01, ***p<0.001; #p<0.05, ##p<0.01, ###p<0.001 (see also **Figure 1—figure supplement 1**).

The online version of this article includes the following source data and figure supplement(s) for figure 1:

**Source data 1.** Raw data shown in **Figure 1** and **Figure 1—figure supplement 1**.

**Figure supplement 1.** Representative temporal patterns of cocaine infusions in vehicle- and ESI-05-treated mice (associated with **Figure 1**).

statistics for this and subsequent results are described in the figure legends). Vehicle-treated mice exhibited a consistently high rate of cocaine infusions with a predominantly regularly spaced pattern over the entire 3 hr session; ESI-05 dose-dependently reduced the rate of cocaine infusions, which was characterized by unevenly spaced intervals and sporadic intake, particularly at the higher ESI-05 dose (10 mg/kg, **Figure 1—figure supplement 1**). Consistent with a previous study (**Slosky et al., 2021**), no significant sex differences were observed for any measure, thus the data were pooled across sexes in subsequent studies.

We next examined the effects of ESI-05 on self-administration across a full range of cocaine doses in mice that acquired stable cocaine self-administration following 10-day training under an FR1 schedule. By using an initial 0.5 mg/kg/infusion followed by descending doses in a single session (1.0, 0.25, 0.125, 0.0625, 0.03125, and 0.0 mg/kg/infusion; see Materials and methods) (**Anderson et al., 2018**), we observed classic inverted U-shaped dose-response curves in C57BL/6J mice, with maximal infusions occurring at the 0.125 mg/kg/infusion dose (**Figure 1E**). After mice achieved stable

self-administration behavior in this multiple-dose paradigm, ESI-05 (3.3 or 10 mg/kg, i.p.) or vehicle (0 mg/kg) was injected 10 min prior to cocaine self-administration. ESI-05 pretreatment led to dose-dependent downward shifts in the cocaine dose-response curve (*Figure 1E*) and an attenuation of cocaine intake (*Figure 1F*).

We next determined whether ESI-05 affected motivation for cocaine self-administration under a PR reinforcement schedule (*Richardson and Roberts, 1996*). Mice first underwent 10 days of cocaine self-administration under an FR1 schedule as described above then were switched to a PR schedule (0.5 mg/kg/infusion). When PR breakpoint stabilized, the effects of ESI-05 (3.3 or 10 mg/kg, i.p.) or vehicle on cocaine infusions and breakpoint were examined. ESI-05 pretreatment dose-dependently reduced cocaine infusions (*Figure 1G*) and breakpoint (*Figure 1H*) compared to vehicle. Thus, ESI-05 reduced the motivation to obtain cocaine when the effort required is progressively increased. Taken together, the above results suggest that ESI-05 reduces cocaine intake and/or motivation for cocaine taking.

## Dopamine neuron-specific knockout of Epac2 impaired cocaine self-administration

We next determined whether conditional deletion of *Epac2* from midbrain dopamine neurons affects cocaine self-administration. Dopamine neuron-specific Epac2 knockout mice (Epac2-cKO) were generated by crossing homozygous *Epac2$^{loxP/loxP}$* mice with heterozygous *Slc6a3$^{IREScre/+}$* mice. Epac2-cKO mice (*Slc6a3$^{IREScre/+}$/Epac2$^{loxP/loxP}$*) and WT controls (*Slc6a3$^{IREScre/+}$/Epac2$^{wt/wt}$* or *Slc6a3$^{+/+}$/Epac2$^{loxP/loxP}$*) were grossly normal in terms of survival, physical appearance, body weight (*Figure 2—figure supplement 1A*), and baseline locomotor activity (*Figure 2—figure supplement 1B*). We verified the selective knockout of *Epac2* from VTA dopamine neurons by single-cell reverse transcriptase-PCR (scRT-PCR). Cell-attached recordings of action potential firing were made in VTA neurons, followed by whole-cell recordings and collection of cytoplasmic contents for scRT-PCR. Neuronal gene expression profiles were characterized by detection of mRNA for *Epac1*, *Epac2*, tyrosine hydroxylase (*Th*, dopamine neuron marker), glutamate decarboxylase 2 (*Gad2*, GABA neuron marker), Solute carrier family 17 (vesicular glutamate transporter), member 6 (*Slc17a6*, glutamate neuron marker), and the housekeeping gene glyceraldehyde-3-phosphate dehydrogenase (*Gapdh*). We found that *Epac2* was expressed in both *Th$^+$* and *Gad2$^+$* neurons in the VTA of WT mice (*Figure 2—figure supplement 2A*) but was only detected in *Gad2$^+$* neurons and not in *Th$^+$* neurons in the VTA of Epac2-cKO mice (*Figure 2—figure supplement 2B*). *Epac1* was rarely expressed in neurons of WT and Epac2-cKO mice, consistent with low expression of *Epac1* in the brain (*Kawasaki et al., 1998*). We also compared the electrophysiological properties of VTA dopamine neurons from WT and Epac2-cKO mice and found no significance differences in the frequency of action potential firing or membrane capacitance (*Figure 2—figure supplement 3*).

Next, we investigated whether cocaine self-administration was altered in Epac2-cKO mice. Epac2-cKO mice and WT controls were trained to self-administer cocaine on an FR1 schedule as described above. There was no significant difference between the percentage of mice in each genotype that acquired stable cocaine self-administration after 10 sessions, with 90.9% (10 of 11) of WT and 84.6% (11 of 13) of Epac2-cKO mice with patent catheters meeting the criteria for stable self-administration ($\chi^2$=0.024, p=0.877). Of the mice that did not meet criteria, one WT mouse and one Epac2-cKO mouse displayed <20 cocaine infusions, while another Epac2-cKO mouse displayed a high numbers of cocaine infusions (>50) but failed to develop active vs. inactive response discrimination. Of the mice that acquired stable self-administration, the average number of training sessions to reach criteria was not significantly different between genotypes (WT, 5.4±0.5 sessions; Epac2-cKO, 6.8±0.7 sessions; $t_{19}$=1.6, p=0.130). However, Epac2-cKO mice showed significant decreases in active nose pokes (*Figure 2A*) and cocaine infusions (*Figure 2B*). WT mice typically displayed a high rate of responding at predominantly regular intervals, while Epac2-cKO mice displayed a pattern resembling that of ESI-05-treated mice characterized by sporadic intake interspersed by long gaps (*Figure 2—figure supplement 4*). Inactive nose pokes were not significantly different between genotypes (*Figure 2C*). Thus, although WT and Epac2-cKO mice similarly acquired stable cocaine self-administration, Epac2-cKO mice exhibited significantly reduced cocaine intake.

To test the hypothesis that Epac2-cKO depresses the reinforcing effects of cocaine, we examined cocaine self-administration in a multiple-dose paradigm using the same cohort of WT and Epac2-cKO

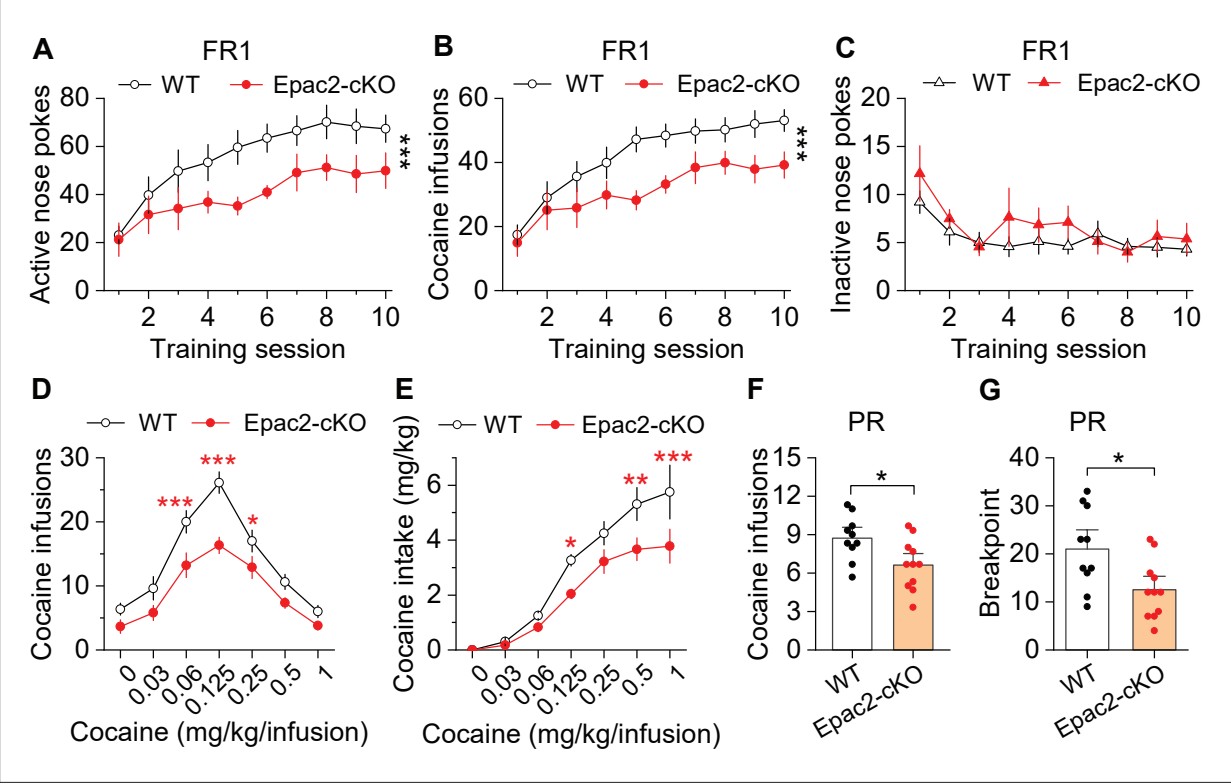

**Figure 2.** Dopamine neuron-specific knockout of Epac2 impaired cocaine self-administration acquisition and reinforcing efficacy. (**A**) Epac2 conditional knockout (Epac2-cKO) mice displayed significantly lower active nose pokes during fixed ratio-1 (FR1) self-administration training (two-way repeated-measures ANOVA: genotype, $F_{1,19}$ = 8.2, p=0.010; training session, $F_{9,171}$ = 10.0, p<0.001; genotype × training session interaction, $F_{9,171}$ = 0.7, p=0.670; n=10–11, 5 male and 5–6 female per group). (**B**) The number of cocaine infusions during training was significantly reduced by Epac2-cKO (two-way repeated-measures ANOVA: genotype, $F_{1,19}$ = 8.3, p=0.010; training session, $F_{9,171}$ = 13.3, p<0.001; genotype × training session interaction, $F_{9,171}$ = 0.9, p=0.543; n=10–11). (**C**) There was no significant difference in inactive nose pokes between genotypes (two-way repeated-measures ANOVA: genotype, $F_{1,19}$ = 1.4, p=0.247; n=10–11, 5 male and 5–6 female per group). (**D–E**) Epac2-cKO mice exhibited a downward shift in the cocaine dose-response curve ( two-way repeated-measures ANOVA: genotype, $F_{1,15}$ = 20.0, p<0.001; cocaine dose, $F_{6,90}$ = 49.4, p<0.001; genotype × cocaine dose interaction, $F_{6,90}$ = 2.2, p=0.046; n=8–9, 4–5 male and 4 female per group) and decreased cocaine intake (two-way repeated-measures ANOVA: genotype, $F_{1,15}$ = 10.4, p=0.006; cocaine dose, $F_{6,90}$ = 62.0, p<0.001; genotype × cocaine dose interaction, $F_{6,90}$ = 2.5, p=0.025; n=8–9, 4 male and 4–5 female per group). (**F–G**) Epac2-cKO mice displayed a reduced number of cocaine infusions (t-test: $t_{19}$=2.6, p=0.019) and lower breakpoint (t-test: $t_{19}$=2.7, p=0.015) under a progressive ratio (PR) reinforcement schedule (n=9–11, 5 male and 4–6 female per group). *p<0.05, **p<0.01, ***p<0.001 (see also ***Figure 2—figure supplements 1–4***).

The online version of this article includes the following source data and figure supplement(s) for figure 2:

**Source data 1.** Raw data shown in ***Figure 2***, ***Figure 2—figure supplement 1***, ***Figure 2—figure supplement 3***, and ***Figure 2—figure supplement 4***.

**Figure supplement 1.** There were no significant differences in body weight and baseline locomotor activity between Epac2 conditional knockout (Epac2-cKO) and wild-type (WT) mice (associated with ***Figure 2***).

**Figure supplement 2.** *Epac2* was selectively deleted from midbrain dopamine neurons in Epac2 conditional knockout (Epac2-cKO) mice (associated with ***Figure 2***).

**Figure supplement 2—source data 1.** Uncropped gels shown in ***Figure 2—figure supplement 2***.

**Figure supplement 3.** Epac2 conditional knockout (Epac2-cKO) did not alter the frequency of spontaneous action potential firing of ventral tegmental area (VTA) dopamine neurons (associated with ***Figure 2***).

**Figure supplement 4.** Representative temporal patterns of cocaine infusions in wild-type (WT) and Epac2 conditional knockout (Epac2-cKO) mice (associated with ***Figure 2***).

mice described in ***Figure 2A–C***. Epac2-cKO mice displayed a significant downward shift in the cocaine dose-response curve compared with WT mice (***Figure 2D***). Epac2-cKO mice also exhibited lower cocaine intake than WT mice (***Figure 2E***). Finally, we examined whether Epac2-cKO mice had altered incentive motivation to obtain cocaine reward under a PR reinforcement schedule in a separate cohort of mice. WT and Epac2-cKO mice underwent 10 days of FR1 cocaine self-administration were then

switched to a PR schedule as described above. Epac2-cKO mice acquired significantly fewer cocaine infusions (*Figure 2F*) and had a significantly reduced breakpoint (*Figure 2G*). Together, these results suggest that Epac2-cKO mice have reduced motivation to acquire cocaine across multiple cocaine doses and when the effort required is progressively increased.

## Epac2 inhibition or Epac2-cKO did not significantly affect oral sucrose self-administration

Does Epac2 activity contribute specifically to drug reinforcement, or is it generalizable to other non-drug rewards? To test this, we examined the effect of systemic Epac2 inhibition and conditional knockout in dopamine neurons on self-administration of an oral sucrose reward under both FR1 and PR schedules. Mice were trained to acquire oral sucrose self-administration under an FR1 schedule (10% sucrose, 35 µl) in 1 hr sessions for 10 days (*Figure 3—figure supplement 1A*). On subsequent days, mice were injected with ESI-05 (3.3 or 10 mg/kg, i.p.) or vehicle 10 min prior to self-administration testing in a Latin Square design. ESI-05 had no significant effect on active nose pokes,

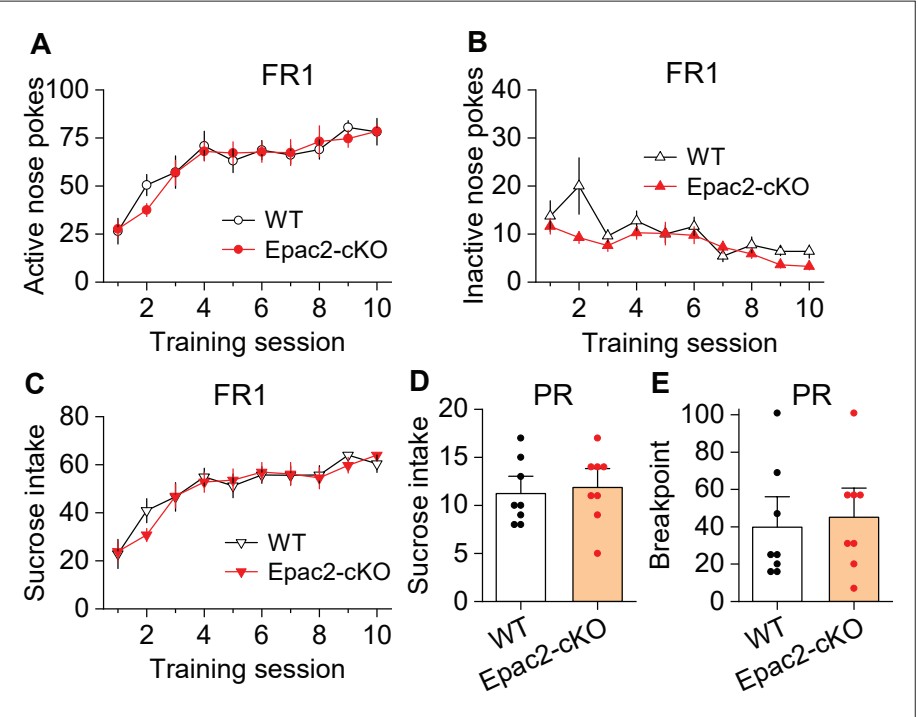

**Figure 3.** Epac2 conditional knockout (Epac2-cKO) did not affect sucrose self-administration under fixed ratio-1 (FR1) and progressive ratio (PR) reinforcement schedules in mice. (**A–C**) Epac2-cKO did not affect the acquisition of sucrose self-administration after 10 days of training under a FR1 schedule (two-way repeated-measures ANOVA: training session on active nose pokes, $F_{9,126} = 20.8$, p<0.001; training session on inactive nose pokes, $F_{9,126} = 6.4$, p<0.001; training session on sucrose intake, $F_{9,126} = 21.3$, p<0.001; genotype on active nose pokes, $F_{1,14} = 0.07$, p=0.802; genotype on inactive nose pokes, $F_{1,14} = 3.1$, p=0.099; genotype on sucrose rewards, $F_{1,14} = 0.08$, p=0.787; training session × genotype interaction in active nose pokes, $F_{9,126} = 0.5$, p=0.849; in inactive nose pokes, $F_{9,126} = 1.5$, p=0.151; in sucrose intake, $F_{9,126} = 0.6$, p=0.8). (**D–E**) Epac2-cKO did not alter the mean number of sucrose intake (t-test: $t_{14}=0.4$, p=0.730) or breakpoints (Mann-Whitney rank: $U=0.6$, p=0.561) under a PR schedule. n=8, 4 male and 4 female per group (see also *Figure 3—figure supplement 1*).

The online version of this article includes the following source data and figure supplement(s) for figure 3:

**Source data 1.** Raw data shown in *Figure 3*, *Figure 3—figure supplement 1*, and *Figure 3—figure supplement 2*.

**Figure supplement 1.** ESI-05 did not affect sucrose self-administration under fixed ratio-1 (FR1) and progressive ratio (PR) reinforcement schedules in mice (associated with *Figure 3*).

**Figure supplement 2.** Representative temporal patterns of sucrose intake in wild-type (WT) and Epac2 conditional knockout (Epac2-cKO) mice (associated with *Figure 3*).

inactive nose pokes, or sucrose intake (*Figure 3—figure supplement 1B-D*). A separate group of mice were trained to self-administer sucrose under a PR reinforcement schedule. Again, ESI-05 did not affect sucrose intake or breakpoint (*Figure 3—figure supplement 1E-F*) under this reinforcement schedule.

We next examined whether Epac2-cKO altered oral sucrose self-administration under FR1 and PR reinforcement schedules. Under an FR1 schedule, there was no significant effect of genotype on active nose pokes, inactive nose pokes, or sucrose intake across the 10-day session (*Figure 3A–C*). Both WT and Epac2-cKO mice displayed a predominantly regularly spaced pattern of sucrose intake over the entire 1 hr session, although bursts of high rates of sucrose intake could also be found in both genotypes (*Figure 3—figure supplement 2*). Similarly, Epac2-cKO did not affect sucrose intake or breakpoint under a PR schedule (*Figure 3D–E*). Taken together, these results suggest that Epac2 inhibition and deletion did not significantly alter the reinforcing effects of a palatable non-drug reward.

## Epac2-cKO led to decreased dopamine release in the striatum in vitro

What might be the mechanisms by which Epac2 in dopamine neurons regulates cocaine self-administration? Drugs of abuse including cocaine increase dopamine levels in the NAc, which is thought to underlie their reinforcing effects (*Di Chiara and Imperato, 1988*; *Koob and Bloom, 1988*). Epac2 interacts with the active zone proteins RIM and Munc13 to facilitate glutamate exocytosis from excitatory synapses (*Ozaki et al., 2000*; *Tengholm and Gylfe, 2017*). RIM conditional knockout from midbrain dopamine neurons abolished action potential-triggered dopamine release (*Liu et al., 2018a*). One possibility is that Epac2 disruption attenuates dopamine release from axon terminals. To test this, we used fast scan cyclic voltammetry (FSCV) to monitor electrically evoked dopamine release in striatal slices from WT and Epac2-cKO mice. FSCV recordings were made in the NAc core (site 1) and shell (site 2), dorsomedial striatum (DMS, site 3), and dorsolateral striatum (DLS, site 4; *Figure 4A and D*). The dorsal striatum is preferentially innervated by dopamine neurons in the substantia nigra pars compacta (SNc), while the NAc is preferentially innervated by VTA dopamine neurons (*Ogawa and Watabe Uchida, 2018*). A fixed stimulation intensity (250 μA, 0.2 ms duration) with either a single electrical stimulus or a train of stimuli (5 pulses, 100 Hz) was applied to evoke dopamine release. The peak amplitudes of evoked dopamine release in response to single pulse and high-frequency burst stimulation were significantly decreased across all striatum subregions sampled in slices from Epac2-cKO mice relative to WT mice (*Figure 4A, B, D and E*). Dopamine reuptake by the dopamine transporter (DAT) is the primary mechanism responsible for extracellular dopamine clearance in the brain (*Gainetdinov et al., 1998*; *Benoit-Marand et al., 2000*). The exponential decay constant $\tau$ of evoked dopamine responses provides a reliable measure for monitoring changes in dopamine reuptake (*Yorgason et al., 2011*). We found that there were no significant changes in $\tau$ from either single pulse or high-frequency burst stimulation across all striatum subregions sampled in slices from Epac2-cKO mice relative to WT mice (*Figure 4A, C, D and F*). Thus, Epac2-cKO mice had decreased evoked dopamine release without significantly altering dopamine reuptake.

## Epac and PKA activation independently enhanced dopamine release in the NAc in vitro

To directly test the effects of Epac2 activation on dopamine release in the NAc shell, we monitored dopamine release evoked by a single electrical stimulus that was applied at 2 min intervals. Sp-8-BnT-cAMPS ('S-220') is an Epac2-selective agonist with specificity for Epac2 over PKA and Epac1 (*Schwede et al., 2015*). Bath application of S-220 led to a concentration-dependent increase in evoked dopamine release in WT slices (*Figure 5A–B*), which was blocked by Epac2-cKO (*Figure 5C*) and ESI-05 (20 μM, *Figure 5D*). In contrast, the PKA inhibitor H89 (1 μM) had no effect (*Figure 5E*). Finally, we examined whether PKA activation altered dopamine release in the NAc. 6-Bnz-cAMP is a membrane permeable and selective PKA activator and does not activate Epac (*Christensen et al., 2003*; *Hewer et al., 2011*). Bath application of 6-Bnz-cAMP (1 μM) increased dopamine release in the NAc shell in slices from WT mice (*Figure 5F*). This effect was blocked by H89 but was not altered by Epac-cKO (*Figure 5F*). Thus, activation of Epac2 and PKA pathways each independently increases evoked dopamine release.

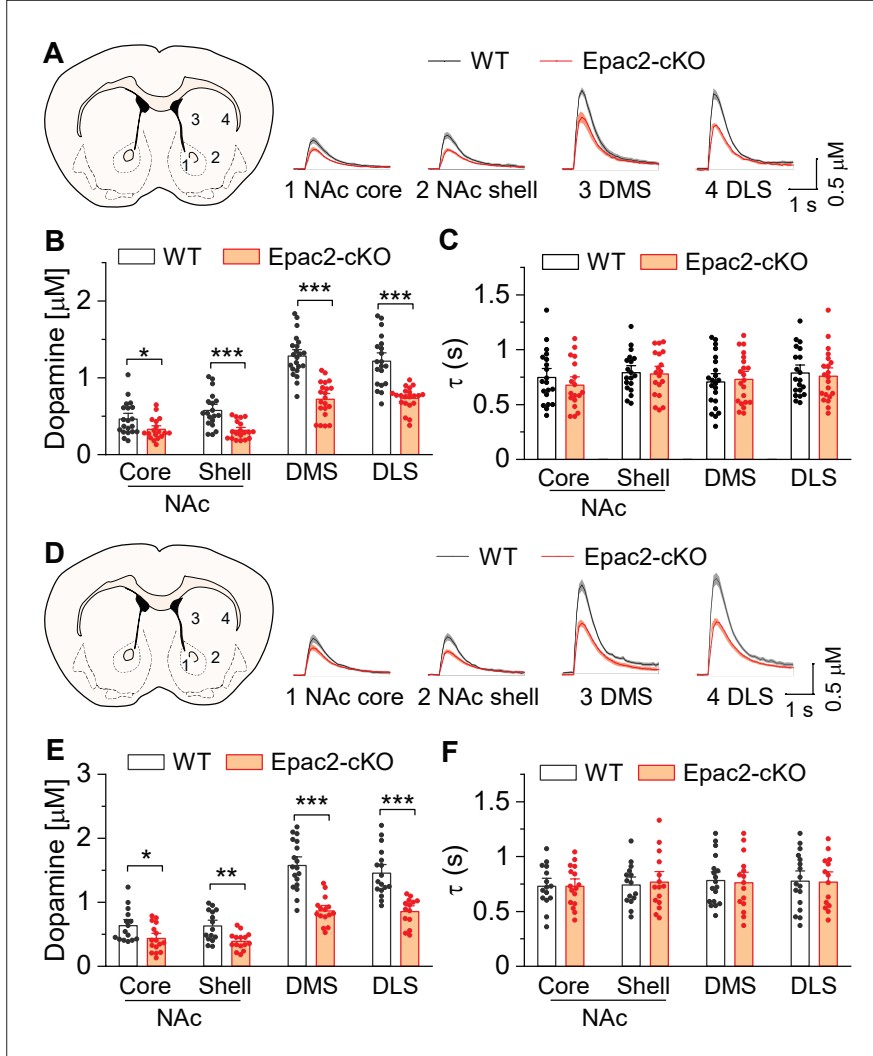

**Figure 4.** Epac2 knockout decreased evoked dopamine release in the striatum in vitro. (**A**) Averaged dopamine concentration vs. time traces from four different striatal subregions (nucleus accumbens [NAc] core, NAc shell, dorsal medial striatum [DMS] and dorsal lateral striatum [DLS]) evoked by a single electrical pulse. (**B**) Compared to wild-type (WT) mice, Epac2 conditional knockout (Epac2-cKO) mice displayed decreased evoked dopamine release across all striatum subregions in response to single pulse stimulation (t-tests: NAc core, $t_{36}$=2.3, p=0.028; NAc shell, $t_{36}$=4.4, p<0.001; DMS, $t_{41}$=7.4, p<0.001; DLS, $t_{39}$=6.1, p<0.001; n=18–22 slices from 4 to 5 male and female mice). (**C**) There were no significant differences in the decay time constants between WT and Epac2-cKO mice with single pulse stimulation across all recording sites (t-tests: NAc core, $t_{36}$=0.9, p=0.366; NAc shell, $t_{36}$=0.2, p=0.842; DMS, $t_{41}$=0.3, p=0.736; DLS, $t_{39}$=0.4, p=0.676; n=18–22 slices from 4 to 5 male and female mice). (**D**) Averaged dopamine concentration vs. time traces from four different striatal subregions in response to 100 Hz of 5-pulse electrical stimulation. (**E**) Compared to WT mice, Epac2-cKO mice displayed decreased evoked dopamine release across all striatum subregions in response to 5-pulse electrical stimulation (t-tests: NAc core, $t_{29}$=2.4, p=0.025; NAc shell, $t_{28}$=3.4, p=0.002; DMS, $t_{31}$=6.3, p<0.001; DLS, $t_{29}$=5.3, p<0.001; n=14–18 slices from 4 to 5 male and female mice). (**F**) There were no significant differences in the decay time constants between WT and Epac2-cKO mice with 5-pulse electrical stimulation (t-tests: NAc core, $t_{29}$=0.006, p=0.999; NAc shell, $t_{28}$=0.3, p=0.770; DMS, $t_{31}$=0.3, p=0.801; DLS, $t_{29}$=0.1, p=0.918; n=14–18 slices from 4 to 5 male and female mice). *p<0.05, **p<0.01, ***p<0.001.

The online version of this article includes the following source data for figure 4:

**Source data 1.** Raw data shown in *Figure 4*.

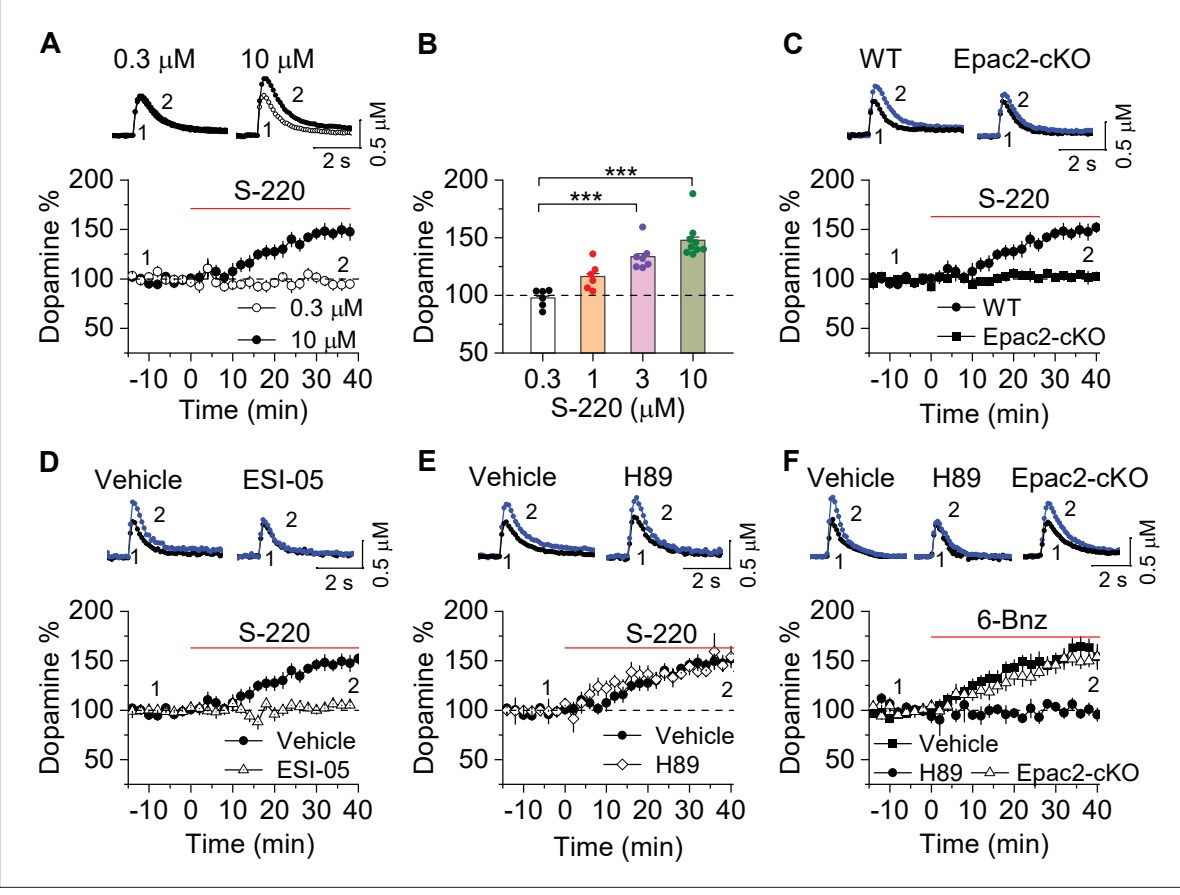

**Figure 5.** Epac2 activation increased dopamine release in the nucleus accumbens (NAc) in vitro. (**A**) Dopamine release was markedly increased by perfusion of S-220 (10 µM) in NAc slices from wild-type (WT) mice (paired *t*-test; $t_8$=9.1, p<0.001; n=9 slices from 3 to 4 male and female mice). (**B**) S-220 increased dopamine release in a dose-dependent manner in NAc slices from WT mice (one-way ANOVA: $F_{3,27}$ = 20.4, p<0.001, n=6–9 slices from 3 to 4 male and female mice). (**C**) S-220 (10 µM) did not increase dopamine release in NAc slices from Epac2 conditional knockout (Epac2-cKO) mice (paired *t*-test: $t_9$ = 0.4, p=0.680, p=0.236; n=9 slices from 3 to 4 male and female mice). (**D**) Preincubation of slices with ESI-05 (20 µM) blocked the S-220-induced increase in dopamine release (*t*-test: $t_{16}$=7.4, p<0.001; n=9 slices from 3 to 4 male and female mice). (**E**) Preincubation of slices with H89 (1 µM) did not alter the S-220-induced increase in dopamine release (*t*-test: $t_{15}$ = 1.47, p=0.162; n=8–9 slices from 3 to 4 male and female mice). (**F**) 6-Bnz (1 µM) increased dopamine release in NAc slices prepared from WT mice (paired *t*-test: $t_8$=6.5, p<0.001; n=9 slices from 3 to 4 male and female mice). The increase was blocked by H89 (1 µM; *t*-test: $t_{14}$ = 5.1, p<0.001; n=7–9 slices from 3 to 4 male and female mice). 6-Bnz also induced an increase in dopamine release in slices from Epac2-cKO mice (paired *t*-test: $t_8$=6.5, p<0.001; n=8 slices from 3 to 4 male and female mice), which was not significantly different from WT mice (*t*-test: $t_{15}$=0.9, p=0.408; n=8–9 slices from 3 to 4 male and female mice). ***p<0.001.

The online version of this article includes the following source data for figure 5:

**Source data 1.** Raw data shown in *Figure 5*.

## Chemogenetic enhancement of dopamine release increased cocaine self-administration in Epac2-cKO mice

If decreased dopamine release resulting from disruption of Epac2 contributes to the impairment of cocaine self-administration, selective enhancement of dopamine release may reverse these behavioral effects. We tested whether chemogenetic enhancement of dopamine release from VTA dopamine neurons can restore cocaine self-administration in Epac2-cKO mice. AAV8-hSyn-DIO-rM3D(Gs)-mCherry or AAV8-hSyn-DIO-mCherry was bilaterally microinjected into the VTA of Epac2-cKO (*Slc6a3*[IREScre/+]/*Epac2*[loxP/loxP]) mice. After 2–3 weeks, transgene expression was verified by immunohistochemistry (*Figure 6A*). rM3D(Gs) and mCherry were expressed in the majority of TH[+] dopamine neurons (85.3 ± 4.2% and 87.1 ± 5.3%, respectively) but were not expressed in TH[-] neurons in the VTA. The AAV expression was predominantly limited to the VTA and was seldom observed in the neighboring SNc. Thus, rM3D(Gs) was selectively expressed in the majority of VTA dopamine neurons.

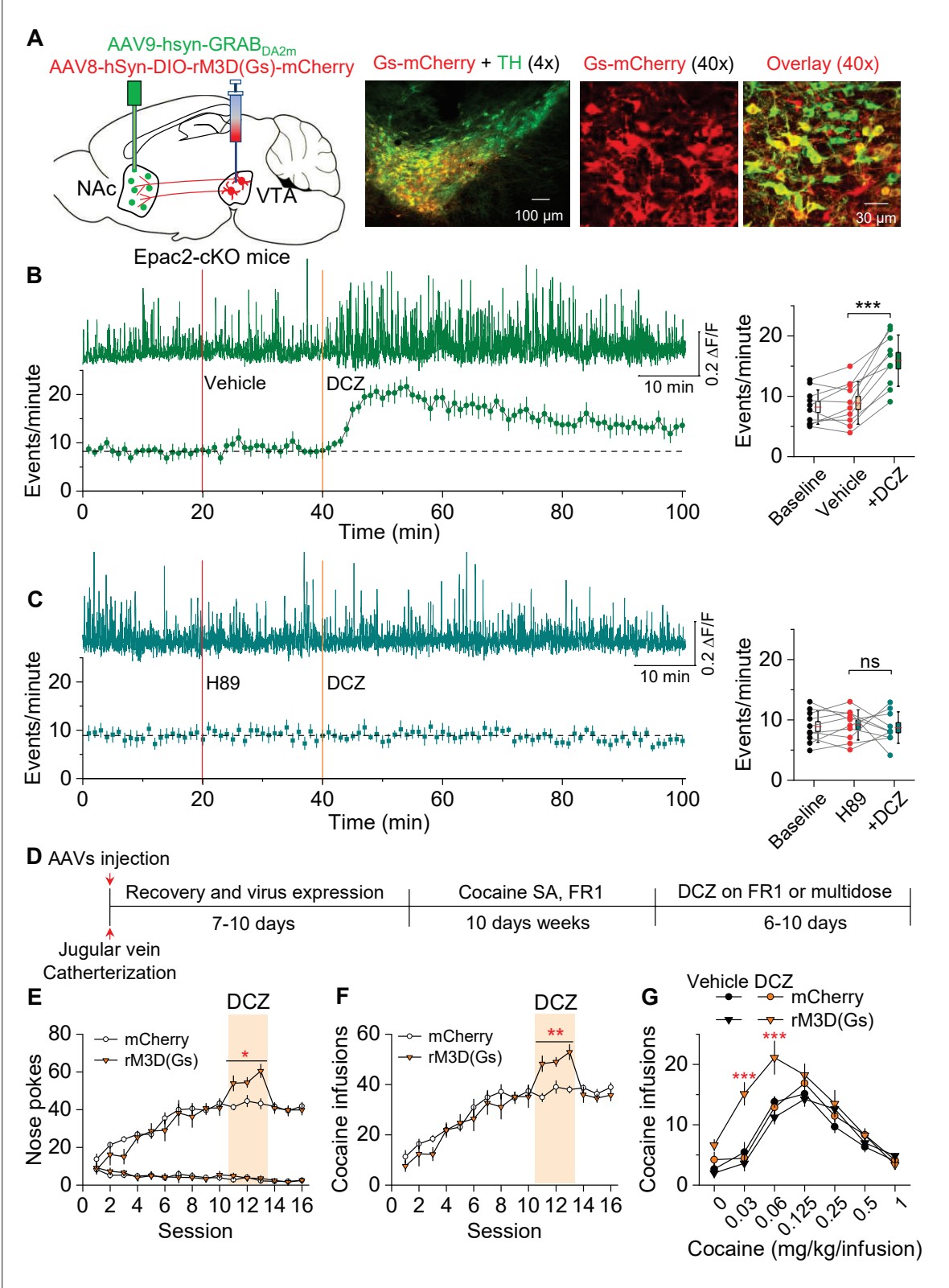

**Figure 6.** Chemogenetic enhancement of dopamine release increased cocaine self-administration in Epac2 conditional knockout (Epac2-cKO) mice. (**A**) AAV8-hSyn-DIO-rM3D(Gs)-mCherry (rM3D(Gs)) or AAV8-hSyn-DIO-mCherry (mCherry) was bilaterally microinjected into the ventral tegmental area (VTA) of Epac2-cKO mice. Immunohistochemistry in VTA sections showed that rM3D(Gs) and mCherry were expressed in the majority of TH[+] dopamine neurons but not in TH[-] neurons. (**B**) Left, representative dopamine signal and the time course of dopamine transients in the nucleus accumbens (NAc)

*Figure 6 continued on next page*

*Figure 6 continued*

of Epac2-cKO mice before and after deschloroclozapine (DCZ). Right, DCZ (10 µg/kg, i.p.) significantly increased the frequency of dopamine transients in the NAc shell of Epac2-cKO mice following vehicle pretreatment (paired *t*-test: $t_9$=7.6, p<0.001; n=10, 5 male and 5 female). (**C**) DCZ did not affect the dopamine transient frequency in mice which received H89 (1 mg/kg, i.p.) pretreatment (paired *t*-test: $t_9$=0.1, p=0.889; n=10, 5 male and 5 female). (**D**) Timeline of AAV injection, cocaine self-administration and DCZ treatment. (**E**) DCZ pretreatment significantly increased active nose pokes (two-way repeated-measures ANOVA: rM3D(Gs), $F_{1,14}$ = 12.8, p=0.003; time, $F_{2,28}$ = 1.9, p=0.164; rM3D(Gs) × time interaction, $F_{2,28}$ = 1.6, p=0.225) without affecting inactive nose pokes (two-way repeated-measures ANOVA: rM3D(Gs), $F_{1,14}$ = 0.3, p=0.613; time, $F_{2,28}$ = 3.0, p=0.065; rM3D(Gs) × time interaction, $F_{2,28}$ = 1.7, p=0.209) in rM3D(Gs)-expressing Epac2-cKO mice under a fixed ratio-1 (FR1) schedule (n=7–9, 4 male and 3–5 female). (**F**) DCZ pretreatment significantly increased the number of cocaine infusions (rM3D(Gs), $F_{1,14}$ = 17.8, p<0.001; time, $F_{2,28}$ = 3.2, p=0.058; rM3D(Gs) × time interaction, $F_{2,28}$ = 1.3, p=0.283; n=7–9, 4 male and 3–5 female) under an FR1 schedule in rM3D(Gs)-expressing Epac2-cKO mice. (**G**) DCZ produced an upward shift in the ascending limb of the cocaine dose-response curve in rM3D(Gs)-expressing Epac2-cKO (two-way repeated-measures ANOVA: rM3D(Gs), $F_{3,30}$ = 8.1, p<0.001; cocaine dose, $F_{6,180}$ = 79.3, p<0.001; rM3D(Gs) × cocaine dose interaction, $F_{18,180}$ = 3.6, p<0.001; n=8–9, 4 male and 4–5 female). ns, p>0.05, *p<0.05, **p<0.01, ***p<0.001 (see also *Figure 6—figure supplement 1*).

The online version of this article includes the following source data and figure supplement(s) for figure 6:

**Source data 1.** Raw data shown in *Figure 6*.

**Figure supplement 1.** Verification of GRAB$_{DA2m}$ injection and activity in the nucleus accumbens (NAc) (associated with *Figure 6*).

In subsequent experiments, we injected Cre-dependent rM3D(Gs)-mCherry into the VTA and a 1:1 mixture of AAV9-hSyn-GRAB$_{DA2m}$ (GRAB$_{DA2m}$) and AAV8-hSyn-mCherry (mCherry) into the NAc shell of Epac2-cKO mice followed by implantation of a 200 µm optical fiber above the GRAB$_{DA2m}$ injection site (*Figure 6—figure supplement 1A*). mCherry was used as a dopamine-independent control for fluorescence changes unrelated to GRAB$_{DA2m}$ activation. After allowing recovery and AAV expression for 2–3 weeks, fiber photometry was performed to detect dopamine dynamics in the NAc. GRAB$_{DA2m}$ sensors are mutant $D_2$ receptors that increase fluorescence intensity with dopamine binding and retain sensitivity to $D_2$ antagonists (*Sun et al., 2020*). We found that the fluorescence transients were abolished by i.p. injection of the $D_2$ antagonist haloperidol (0.1 mg/kg, i.p.; *Figure 6—figure supplement 1B*), indicating that the transients are due to dopamine binding the sensor.

DCZ is a DREADD actuator that shows greater affinity and potency for DREADDs than clozapine N-oxide, is highly brain-penetrant, and is metabolically stable (*Nagai et al., 2020*). In vivo two-photon imaging of somatosensory cortex neurons that co-expressed $G_q$-DREADD (hM3Dq) and GCaMP6 revealed that i.p. injection of DCZ led to a rapid increase in GCaMP6 activity in mice that peaked at about 10 min and plateaued for at least 150 min. Nevertheless, we employed subcutaneous injection of DCZ to provide relatively even and long-lasting activation of Gs- and Gi-DREADDs. Subcutaneous injection of DCZ (100 µg/kg) 20 min after i.p. vehicle pretreatment caused a rapid and robust increase (~30 min) in the frequency of dopamine transients in the NAc shell, which was followed by a more modest and sustained increase lasting for the duration of the fiber photometry recordings (*Figure 6B*). Pretreatment with the PKA inhibitor H89 (1 mg/kg, i.p.) blocked the effect of DCZ to increase dopamine transient frequency (*Figure 6C*). Thus, DCZ induced a PKA-dependent increase in dopamine release in Epac2-cKO mice.

We determined whether rM3D(Gs) activation by DCZ in VTA dopamine neurons can reverse the impairment of cocaine self-administration in Epac2-cKO mice. Epac2-cKO mice received bilateral intra-VTA injections of Cre-dependent rM3D(Gs) or mCherry and underwent jugular vein catheterization for cocaine self-administration. The timeline of the AAV injection and behavioral tests is shown in *Figure 6D*. Mice were trained to acquire stable cocaine (0.5 mg/kg/infusion) self-administration for 10 days under an FR1 schedule. We examined whether enhancing dopamine release through rM3D(Gs) activation altered cocaine intake in mice that established stable cocaine self-administration. Mice received subcutaneous injection of DCZ (100 µg/kg) 10 min prior to the testing of cocaine self-administration for three daily sessions that lasted for 3 hr per session. DCZ increased the number of active nose pokes (*Figure 6E*) and cocaine infusions (*Figure 6F*) in rM3D(Gs)-expressing Epac2-cKO mice. In contrast, DCZ did not significantly affect active nose pokes, inactive nose pokes (*Figure 6E*), or cocaine infusions (*Figure 6F*) in mCherry-expressing Epac2-cKO mice, indicating that the effects of DCZ are mediated by rM3D(Gs) activation. In a multiple-dose cocaine self-administration paradigm, DCZ produced an upward shift of the ascending limb of the inverted U-shaped curve of cocaine self-administration in Epac2-cKO mice that expressed rM3D(Gs) in VTA dopamine neurons but did not alter the cocaine dose-response curve in Epac2-cKO mice that expressed control vector (*Figure 6G*).

These results suggest that rM3D(Gs) activation with DCZ reversed the impairment of cocaine self-administration in the Epac2-cKO mice.

## Chemogenetic inhibition of dopamine release decreased cocaine self-administration in WT mice

We next determined whether selective inhibition of dopamine release from VTA dopamine neurons would mimic the effect of Epac2 disruption on cocaine self-administration in WT mice. WT ($Slc6a3^{IRE-Scre/+}$) mice received bilateral intra-VTA injections of AAV8-hSyn-DIO-hM4D(Gi)-mCherry or mCherry and intra-NAc shell injection of 1:1 mixed GRAB$_{DA2m}$ and AAV-mCherry followed by implantation of a 200 µm optical fiber (**Figure 7A**). After recovery for 2–3 weeks, dopamine release was monitored with in vivo fiber photometry. Subcutaneous injection of DCZ (100 µg/kg) significantly decreased dopamine transient frequency in the NAc shell of mice that expressed hM4D(Gi) but not mCherry (**Figure 7B**). After the completion of the experiments, expression of the AAVs in the VTA and NAc was verified by immunohistochemistry. hM4D(Gi) and mCherry were expressed in the majority of TH$^+$ dopamine neurons (83.8 ± 4.1% and 84.7 ± 3.4%, respectively) but were not expressed in TH$^-$ neurons in the VTA (**Figure 7A**). Thus, hM4D(Gi) was selectively expressed in the majority of VTA dopamine neurons.

Having shown that hM4D(Gi) activation decreased dopamine release, we next investigated its effects on cocaine self-administration in WT mice. The timeline of AAV injection and behavioral tests is shown in **Figure 7C**. Mice were trained to acquire stable cocaine (0.5 mg/kg/infusion) self-administration for 10 days under an FR1 schedule. Mice continued to maintain cocaine self-administration on subsequent days while receiving subcutaneous injections of DCZ (100 µg/kg) 10 min prior to the testing. DCZ significantly decreased the number of active nose pokes and cocaine infusions in hM4D(Gi)-expressing WT mice, but did not significantly affect active nose pokes, cocaine infusions, or inactive nose pokes in mCherry-expressing WT mice (**Figure 7D–E**). In a multiple-dose cocaine self-administration paradigm, DCZ produced a downward shift of the cocaine dose-response curve in WT mice that expressed hM4D(Gi) in VTA dopamine neurons but did not alter the cocaine dose-response curve in WT mice that expressed mCherry (**Figure 7F**). These results indicate that hM4D(Gi)-mediated inhibition of dopamine release decreases cocaine self-administration and mimics the effect of Epac2 disruption.

## Discussion

Here, we show that Epac2 inhibition or genetic deletion in midbrain dopamine neurons decreased cocaine self-administration across multiple cocaine doses and reinforcement schedules. Epac2-cKO led to decreased evoked dopamine release, whereas Epac2 agonism enhanced dopamine release in the NAc. To determine whether the attenuated dopamine response contributes to the impairment of cocaine self-administration, we expressed rM3D(Gs) or hM4D(Gi) selectively in VTA dopamine neurons and found that enhancing dopamine release with rM3D(Gs) activation reversed the reduction of cocaine self-administration in Epac2-cKO mice, while reducing dopamine release with hM4D(Gi) activation attenuated cocaine self-administration in WT mice. A hypothetical model that summarized these results is shown in **Figure 8**. Taken together, these results reveal a critical function of Epac2 in modulating dopamine release and cocaine reinforcement.

ESI-05 caused a dose-dependent decrease in cocaine intake in WT mice under an FR1 reinforcement schedule. In a multiple-dose self-administration paradigm, ESI-05 dose-dependently decreased cocaine infusions and intake at cocaine doses that fall on both the ascending and descending limbs of the inverted U-shaped dose-response curve. Similarly, although Epac2-cKO and WT mice required a similar number of training sessions to reach stable cocaine self-administration, Epac2-cKO mice displayed an overall reduction in self-administration across low and high cocaine unit doses. This depression of the dose-response curve suggests a general decrease in motivation to self-administer cocaine rather than a simple shift in the potency of cocaine (*Mello and Negus, 1996*). In support of this, Epac2 inhibition or genetic deletion also decreased active nose pokes, cocaine infusions, and breakpoint under PR reinforcement. ESI-05 and Epac2-cKO did not alter inactive nose pokes, nor did they alter oral sucrose self-administration under FR1 or PR reinforcement schedules. Thus, the reduction in cocaine self-administration was unlikely to be caused by a general suppression of operant

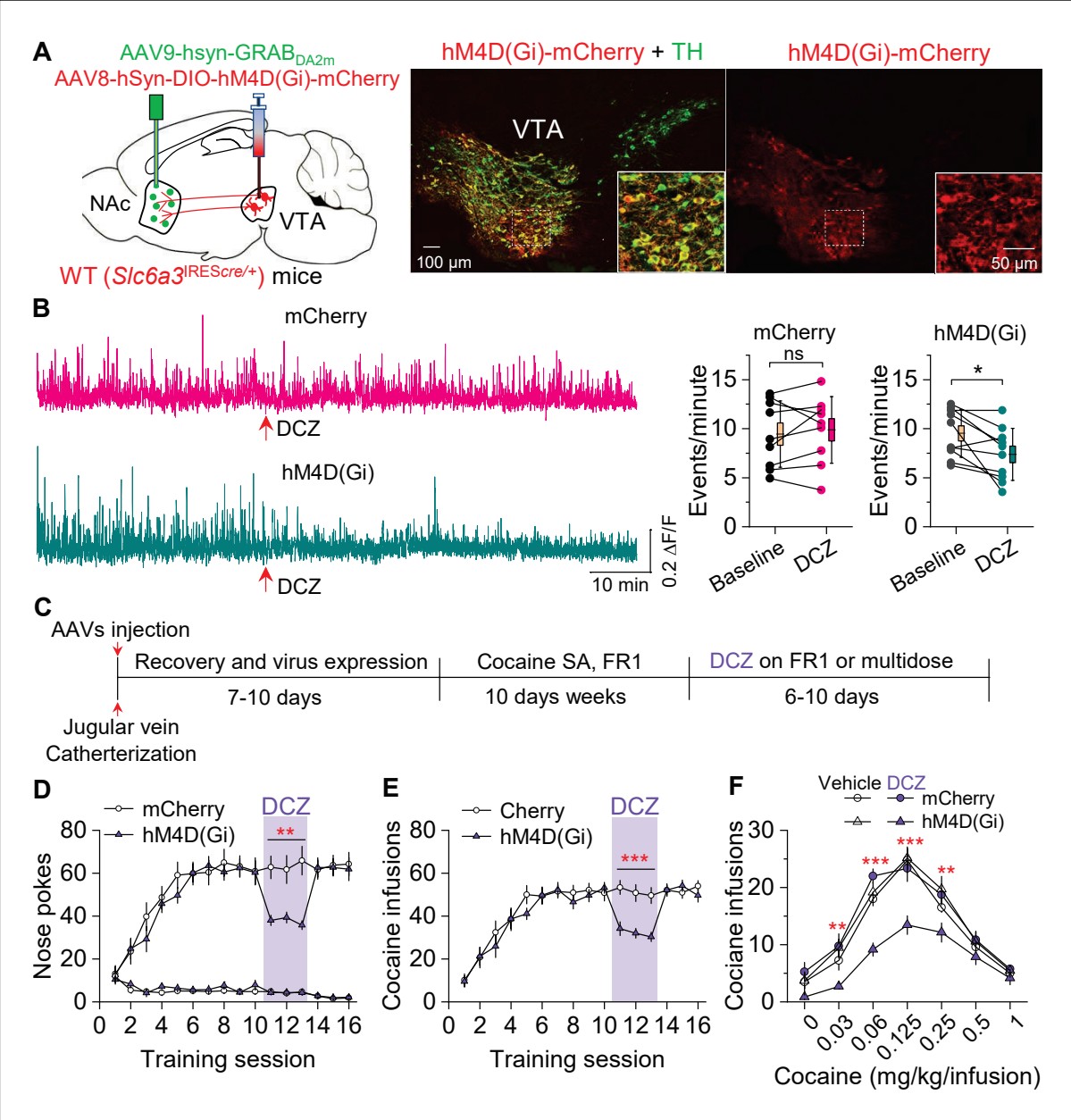

**Figure 7.** Chemogenetic inhibition of dopamine release decreased cocaine self-administration in wild-type (WT) mice. (**A**) AAV8-hSyn-DIO-hM4D(Gi)-mCherry (hM4D(Gi)) or AAV8-hSyn-DIO-mCherry (mCherry) was bilaterally microinjected into the ventral tegmental area (VTA) of WT mice, and the dopamine sensor AAV9-hsyn-DA2m (GRAB$_{DA2m}$) was injected into the NAc shell followed by implantation of a fiber-optic cannula. Immunohistochemistry showed that hM4D(Gi) was expressed in the majority of TH$^+$ dopamine neurons (green) but was not expressed in TH$^-$ neurons in the VTA. (**B**) Left, representative dopamine signal in the nucleus accumbens (NAc) of WT mice before and after deschloroclozapine (DCZ) treatment. Right, DCZ (10 µg/kg, i.p.) did not alter the frequency of dopamine transients in mice with mCherry expression (paired $t$-test: $t_8$=0.7, p=0.514; n=9, 5 male and 4 female). The frequency of dopamine transients was significantly decreased after DCZ treatment in WT mice expressing hM4D(Gi) in VTA dopamine neurons (paired $t$-test: $t_9$=2.9, p=0.017; n=10, 5 male and 5 female). (**C**) Timeline of cocaine self-administration and DCZ treatment. (**D–E**) DCZ pretreatment significantly decreased active nose pokes (two-way repeated-measures ANOVA: hM4D(Gi), $F_{1,15}$ = 18.6, p<0.001; time, $F_{2,30}$ = 0.03, p=0.966; hM4D(Gi) × time interaction, $F_{2,30}$ = 2.5, p=0.096) and cocaine infusions (two-way repeated-measures ANOVA: hM4D(Gi), $F_{1,15}$ = 25.7, p<0.001; time, $F_{2,30}$ = 2.2, p=0.132; hM4D(Gi) × time interaction, $F_{2,30}$ = 0.02, p=0.986) without altering inactive nose pokes (hM4D(Gi), $F_{1,15}$ = 0.01, p=0.911; time, $F_{2,30}$ = 0.5, p=0.627; hM4D(Gi) × time interaction, $F_{2,30}$ = 0.9, p=0.402) in hM4D(Gi)-expressing WT mice (n=8–9, 4 male and 4–5 female). (**F**) DCZ produced a significant downward shift of the cocaine dose-response curve (two-way repeated-measures ANOVA: hM4D(Gi), $F_{3,26}$ = 13.8, p<0.001; cocaine dose, $F_{6,156}$ = 100.2, p<0.001; hM4D(Gi) × cocaine dose interaction, $F_{18,156}$ = 2.2, p=0.006; n=7–8, 4 male and 3–4 female) in WT mice that expressed hM4D(Gi) in VTA dopamine neurons. ns, p>0.05, *p<0.05, **p<0.01, ***p<0.001.

*Figure 7 continued on next page*

*Figure 7 continued*

The online version of this article includes the following source data for figure 7:

**Source data 1.** Raw data shown in *Figure 7*.

responding. Together, these studies demonstrate that Epac2 in midbrain dopamine neurons is a critical contributor to cocaine reinforcement across multiple self-administration paradigms.

Cocaine and sucrose increase dopamine levels by reducing its uptake and increasing its release, respectively (*Anderson and Pierce, 2005*; *Patriarchi et al., 2018*). It was somewhat surprising that ESI-05 and Epac2 attenuated cocaine self-administration but did not significantly alter sucrose self-administration. However, similar differential modulation of cocaine vs. sucrose self-administration by pharmacological agents has been reported (*Levy et al., 2007*; *Romieu et al., 2008*; *Mu et al., 2021*). Although there is significant overlap in the neuronal circuits encoding food and drug rewards (*Volkow et al., 2012*), the majority of NAc neurons exhibited different firing patterns in response to food and cocaine rewards (*Carelli, 2002*). As mentioned in the Introduction, repeated cocaine exposure leads to decreased $G_{\alpha i/o}$ expression in the VTA and an upregulation of cAMP signaling (*Nestler et al., 1990*; *Striplin and Kalivas, 1992*). It is likely that Epac2 is preferentially activated following chronic cocaine self-administration, which might explain why Epac2 disruption attenuated cocaine self-administration but not sucrose self-administration.

Epac agonism was shown to facilitate vesicle exocytosis by interacting with the active zone proteins RIM and Munc13 (*Ozaki et al., 2000*; *Tengholm and Gylfe, 2017*). RIM is clustered in dopamine axons, and conditional knockout of RIM from midbrain dopamine neurons abolished action potential-triggered dopamine release (*Liu and Kaeser, 2019*). We tested the hypothesis that Epac2 modulates axonal dopamine release in the striatum. Using FSCV to record evoked dopamine release in brain slices, we found that an Epac2-selective agonist dose-dependently increased evoked dopamine

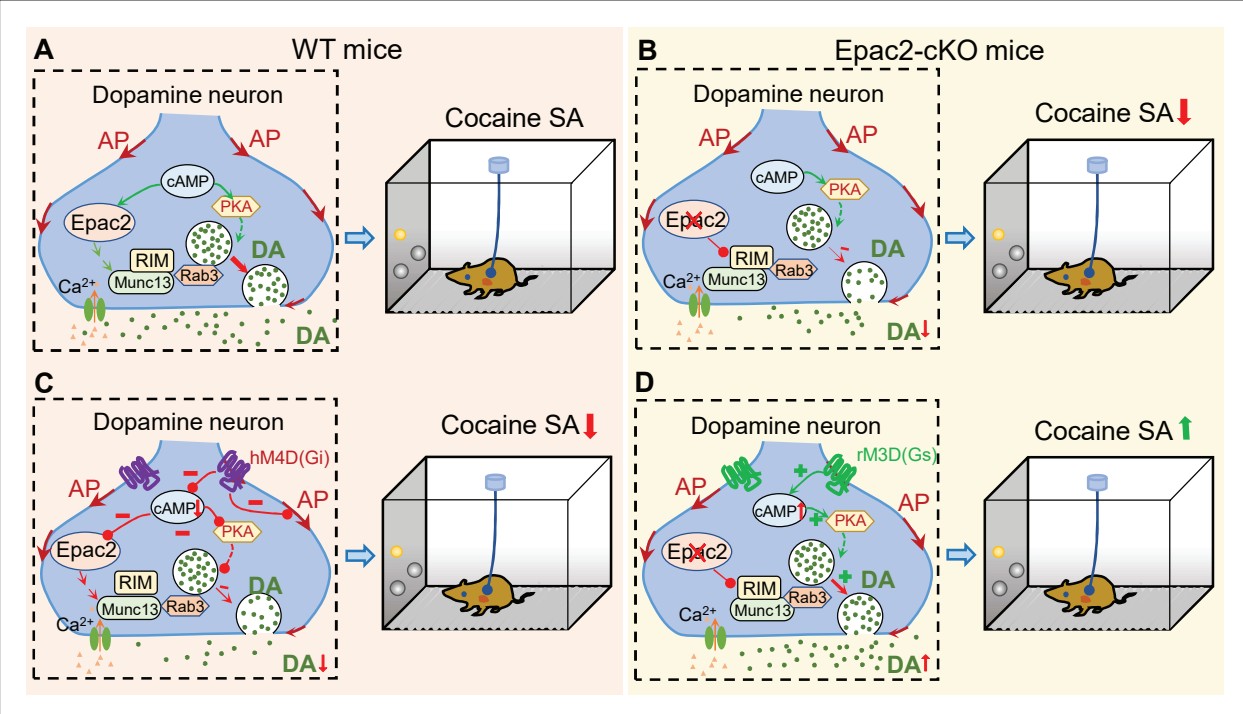

**Figure 8.** A hypothetical model that explains the relationship between Epac2 disruption, decreased dopamine release, and impairment of cocaine self-administration. (**A**) Epac2 is known to interact with the active zone proteins RIM, Munc13, and Rab3. A potential interaction of Epac2 with active zone proteins in the axonal varicosity of dopamine neurons facilitates dopamine exocytosis, which supports cocaine reinforcement during self-administration (SA). (**B**) Epac2 conditional knockout (Epac2-cKO) mice led to decreased dopamine release and cocaine SA. (**C**) hM4D(Gi)-mediated inhibition of action potential (AP) firing and dopamine release from axonal terminals of ventral tegmental area (VTA) dopamine neurons in WT mice led to a reduction of cocaine self-administration (mimicking the effect of Epac2-cKO). (**D**) Enhancing dopamine release via rM3D(Gs) activation in VTA dopamine neurons increases dopamine release via the PKA pathway and enhances cocaine self-administration in Epac2-cKO mice.

release in the NAc. This effect was abolished by Epac2 inhibition and *Epac2* genetic deletion but was unaffected by PKA inhibition. Further, Epac2-cKO mice had attenuated evoked dopamine release in vitro across the NAc and dorsal striatum. RIM mediates vesicle tethering and is required for $Ca^{2+}$-channel recruitment to release sites (*Acuna et al., 2016*). Epac2 may therefore facilitate dopamine exocytosis from midbrain dopamine neurons by interacting with RIM and other active zone proteins and prompting $Ca^{2+}$-dependent dopamine release.

There is a possibility that the Epac2 agonist S-220 affected other neurotransmitter systems to indirectly modulate dopamine release. Optogenetic stimulation of cholinergic interneurons in the NAc induced dopamine release via activation of nicotinic acetylcholine receptors (nAChRs) (*Cachope et al., 2012*; *Threlfell et al., 2012*). GABA tonically depresses axonal dopamine release in the dorsal striatum via $GABA_A$ and $GABA_B$ receptors (*Roberts et al., 2020*). Thus, acetylcholine and GABA have indirect but major impacts on dopamine release. However, the S-220-induced enhancement of dopamine release was abolished by Epac2-cKO from dopamine neurons, suggesting that dopamine axonal terminals are the predominant site for the action of S-220. In future studies, it would be interesting to test whether the S-220-induced enhancement of dopamine release is altered by nAChR and GABA receptor blockers.

Drugs of abuse including cocaine increase dopamine levels in the NAc, which is thought to underlie their reinforcing effects (*Anderson and Pierce, 2005*; *Volkow et al., 2006*). As disruption of Epac2 leads to reduced synaptically released dopamine, it is likely that cocaine-mediated DAT inhibition in mice with disrupted Epac2 would cause a lesser increase in extracellular dopamine levels and attenuated dopamine-driven behaviors. To test whether decreased dopamine release contributes to the impairment of cocaine self-administration, we expressed rM3D(Gs) in VTA dopamine neurons in Epac2-cKO mice and found that DCZ increased the frequency dopamine transients by activating PKA and increased cocaine self-administration under FR1 reinforcement. In the multiple-dose self-administration paradigm, rM3D(Gs) activation induced an upward shift of the ascending limb of the cocaine dose-response curve without affecting the descending limb. The increase in operant responding on the ascending limb could be attributable to an increase in the reinforcing efficacy of cocaine (*Mello and Negus, 1996*; *Caine et al., 2002*; *Roberts et al., 2013*). Since rM3D(Gs) activation increases dopamine release, cocaine infusions at low unit doses likely result in greater dopamine levels, which enhances reinforcement and promotes operant responding (*Mello and Negus, 1996*; *Caine et al., 2002*; *Roberts et al., 2013*). The DCZ-induced increase in the reinforcing efficacy of cocaine may counter the rate-decreasing effects of higher cocaine unit doses (*Zimmer et al., 2011*), which may explain why rM3D(Gs) activation did not significantly shift the descending limb. Together, these experiments demonstrate that the impairment of cocaine self-administration caused by Epac2 disruption is largely reversible with restoration of midbrain dopamine release.

In contrast, the activation of hM4D(Gi) in WT (*Slc6a3*^IREScre/+^/*Epac2*^wt/wt^) mice decreased the frequency of dopamine transients in the NAc in vivo, decreased cocaine self-administration under FR1 reinforcement schedule, and induced an overall downward shift of the cocaine dose-response curve. This phenotype mirrors that of Epac2-cKO mice and indicates a reduction of self-administration across a full range of cocaine unit doses (*Mello and Negus, 1996*). Similarly, chemogenetic inhibition of VTA dopamine neurons via hM4D(Gi) suppressed heroin self-administration under an FR1 reinforcement schedule (*Corre et al., 2018*). Conversely, knockdown of D2 autoreceptors in midbrain dopamine neurons increased dopamine levels due to reduced autoinhibition and caused a general upward shift of the cocaine dose-response curve (*Chen et al., 2018*). Thus, bidirectional modulation of dopamine release is capable of enhancing or depressing operant responding for cocaine across multiple unit doses and does not result in a simple shift in cocaine potency. Further, Gq-DREADD-mediated stimulation of VTA dopamine neurons potentiated cue and cocaine-priming induced reinstatement, whereas Gi-DREADD reduced reinstatement (*Mahler et al., 2018*).

We acknowledge the limitations of our DREADD experiments. First, Gs-DREADD stimulation was used to restore dopamine release in Epac2-cKO mice but did not restore Epac2 function. Further studies could determine whether viral expression of *Epac2* in VTA dopamine neurons in Epac2-cKO mice could restore dopamine release and cocaine self-administration to the level of WT mice. Second, our DREADD studies were limited to experiments that were essential to test the central hypothesis that the Epac2-cKO-induced reduction of dopamine release contributes to the deficit in cocaine self-administration. Both mimicry and occlusion can provide insight into whether two biological processes

occur by the same mechanism. We have shown that hM4D(Gi) inhibition of VTA dopamine neurons in WT mice mimicked the effects of Epac2-cKO on cocaine self-administration. A complementary experiment would be to determine whether hM4D(Gi) inhibition of dopamine neurons further decreases cocaine self-administration in Epac2-cKO mice or if the effect of hM4D(Gi) inhibition is occluded by Epac-cKO.

It is likely that Epac2 regulates behavior via mechanisms that extend beyond the facilitation of presynaptic dopamine release. Epac signals downstream via myriad effectors, including Rap, Ras, Rho, phospholipase C, and mitogen-activated protein kinases (*Laurent et al., 2012*). Epac can be activated by both cAMP-dependent and -independent mechanisms (*Shibasaki et al., 2004*; *Baameur et al., 2016*; *Robichaux and Cheng, 2018*). Repeated cocaine administration induces structural plasticity in the NAc (*Cahill et al., 2016*; *Barrientos et al., 2018*; *Kalivas, 2018*), of which Rap1 GTPase signaling plays a critical role (*Cahill et al., 2016*). Cocaine self-administration and seeking depend on both dopamine-dependent and -independent mechanisms (*Schmidt and Pierce, 2010*; *Mereu et al., 2020*). A prominent function of Epac is to regulate synaptic transmission via both presynaptic and postsynaptic mechanisms. Postsynaptically, Epac2 modulates AMPAR trafficking by inducing changes in AMPAR subunit composition (*Woolfrey et al., 2009*; *Liu et al., 2016*) and enhances open-channel probability of GluA3-AMPARs (*Gutierrez-Castellanos et al., 2017*). Presynaptically, Epac2 facilitates glutamate release and enables long-term potentiation (LTP) (*Fernandes et al., 2015*; *Martín et al., 2020*). Genetic deletion of *Epac1/2* or *Epac2* led to the impairment of LTP in the hippocampus, spatial social interaction, and spatial learning and memory (*Yang et al., 2012*). Addictive drugs such as cocaine induce long-term synaptic plasticity in the reward circuitry of the brain (*Wolf, 2016*; *Dong et al., 2017*; *Wang et al., 2018*; *Zinsmaier et al., 2022*). We have shown that Epac2, but not Epac1, was required for alterations in excitatory and inhibitory transmission in VTA dopamine neurons induced by non-contingent cocaine exposure (*Liu et al., 2016*; *Tong et al., 2017*). It is likely that Epac2 acts via multifactorial mechanisms that together promote cocaine-induced neuroadaptations and drug reinforcement.

## Materials and methods

### Key resources table

| Reagent type (species) or resource | Designation | Source or reference | Identifiers | Additional information |
|---|---|---|---|---|
| Strain, strain background (*Mus musculus*) | C57BL/6J mice | The Jackson Laboratory | Stock#: 000664 | |
| Genetic reagent (*Mus musculus*) | Heterozygous *Slc6a3*[IREScre/+] mice | The Jackson Laboratory | Stock#: 006660 | Maintained on the C57BL/6J background |
| Strain, strain background (*Adeno-associated virus*) | AAV9-hSyn-GRAB$_{DA2m}$ | WZ Biosciences | | |
| Strain, strain background (*Adeno-associated virus*) | AAV8-hSyn-mCherry | Addgene | Catalog#: 114472-AAV8 | |
| Strain, strain background (*Adeno-associated virus*) | AAV8-hSyn-DIO-rM3D(Gs)-mCherry | Addgene | Catalog#: 50458-AAV8 | |
| Strain, strain background (*Adeno-associated virus*) | AAV8-hSyn-DIO-hM4D(Gi)-mCherry | Addgene | Catalog#: 44362-AAV8 | |
| Strain, strain background (*Adeno-associated virus*) | AAV8-hSyn-DIO-mCherry | Addgene | Catalog#: 50459-AAV8 | |
| Antibody | Rabbit polyclonal anti-TH | Santa Cruz Biotechnology | SC-14007, Lot: C2707 | 1:300 |
| Antibody | Goat anti-rabbit IgG Alexa Fluor-488 | Cell Signaling | Stock#: 4412 | 1:300 |
| Other | NeuroTrace 640/660 Deep-Red Fluorescent Nissl Stain | Thermo Fisher Scientific | N-21483 | 1:150 |
| Software, algorithm | Origin 2022b | OriginLab Corporation | | |
| Software, algorithm | RStudio | RStudio, PBC | | DIO Link (Github): GitHub - xiaojieliu17/Fiber-photometry |

## Animals

Animal maintenance and use were in accordance with protocols approved by the Institutional Animal Care and Use Committee of Medical College of Wisconsin. Homozygous *Epac2*-floxed mice (*Epac2$^{loxP/loxP}$*) were generated as described previously (*Pereira et al., 2013*). Heterozygous *Slc6a3*$^{IREScre/+}$ mice (Jax stock#: 006660) were obtained from the Jackson Laboratory (Bar Harbor, ME). All the mouse lines were maintained on the same C57BL/6 background. Dopamine neuron-specific Epac2 knockout mice (Epac2-cKO) were generated by crossing homozygous *Epac2$^{loxP/loxP}$* mice with heterozygous *Slc6a3*$^{IREScre/+}$ mice. We then crossed these compound *Slc6a3*$^{IREScre/+}$/*Epac2$^{loxP/wt}$* mice with *Epac2$^{loxP/loxP}$* mice to create dopamine neuron-specific Epac2-cKO mice (*Slc6a3*$^{IREScre/+}$/*Epac2$^{loxP/loxP}$*). *Slc6a3*$^{IREScre/+}$/*Epac2$^{wt/wt}$* or *Slc6a3$^{+/+}$*/*Epac2$^{loxP/loxP}$* mice were used as the WT group. Genotyping was performed by PCR analysis of ear or tail biopsies.

All animals were given ad libitum access to food and water in a room with controlled temperature (23 ± 1°C) and humidity (40–60%) under a reverse light cycle. Adult mice of either sex (10–12 weeks) were used for experiments. Mice were handled daily for 3–6 days prior to experiments. Mice were single-housed after intravenous catheterization and/or fiber-optic cannula implantation to prevent cage-mates from damaging the implanted catheters and cannulae.

## Stereotaxic surgeries and AAV injections

Mice were anesthetized with ketamine (90 mg/kg, i.p.) and xylazine (10 mg/kg, i.p.) and placed in a robot stereotaxic system (Neurostar, Tübingen, Germany). A volume of AAVs (see below) were delivered through a glass capillary Nanoinjector (Neurostar) or a Nanoject III Programmable Nano-liter Injector (Drummond Scientific Company, Broomall, PA). The injection rate was 60 nl/min and the injector was kept in place for 5 min to ensure adequate diffusion from the injector tip (~30 µm). Mice were kept on a heating pad during surgery and for 2–3 hr following surgery to maintain body temperature. After the surgery, animals received subcutaneous injection of analgesic (buprenorphine-SR, 0.05 mg/kg) and were allowed to recover ~2 weeks before behavioral tests (see below).

For experiments that examined DREADD activation in VTA dopamine neurons on dopamine release in the NAc, AAV8-hSyn-DIO-rM3D(Gs)-mCherry, AAV8-hSyn-DIO-hM4D(Gi)-mCherry, or AAV8-hSyn-DIO-mCherry (200 nl each; Addgene, Watertown, MA) was bilaterally injected into VTA in either WT (*Slc6a3*$^{IREScre/+}$/*Epac2$^{wt/wt}$*) or Epac2-cKO mice (AP, –3.1 mm; ML, ±1.0 mm; DV, –4.8 mm at a 7° angle). Then, 200 nl of 1:1 mixture of AAV9-hSyn-GRAB$_{DA2m}$ and AAV8-hSyn-mCherry was injected into the NAc shell. After the AAV injection, a low autofluorescence fiber-optic cannula (OD = 200 µm, NA = 0.57; Doric Inc, Québec, Canada) was lowered to the NAc just above the AAV injection site. The cannula was secured by two small screws into the skull (00–96 × 1/16; Invivo1) that were attached to the skull by C&B Metabond quick adhesive cement.

For experiments that examined the effects of DREADD activation in VTA dopamine neurons on cocaine self-administration, mice first underwent jugular vein catheterization (see below), then AAV8-hSyn-DIO-rM3D(Gs)-mCherry, AAV8-hSyn-DIO-hM4D(Gi)-mCherry, or AAV8-hSyn-DIO-mCherry was bilaterally injected into VTA in either Epac2-cKO mice or WT (*Slc6a3*$^{IREScre/+}$/*Epac2$^{wt/wt}$*) as described above.

## Jugular vein catheterization

Mice were anesthetized with ketamine (90 mg/kg, i.p.) and xylazine (10 mg/kg, i.p.), and a round-tip polyurethane catheter with a bead 1.2 cm from the catheter tip (#C20PU-MJV1934, Instech Laboratories, Inc, Plymouth Meeting, PA) was inserted into the right jugular vein. The catheter was connected to a vascular access button (25-gauge; #VAM1B/25, Instech) and implanted subcutaneously on the back of the mice. This system decreases trauma to the inner lining of the vessel walls and allows the maintenance of positive pressure, which greatly prolongs the catheter patency and enables group housing (*Slosky et al., 2020*; *Slosky et al., 2021*). Catheters were flushed with 0.1 ml of heparinized saline (30 units/ml) after surgery to prevent coagulation. Catheter patency tests (xylazine, 2 mg/kg, 0.5 ml, i.v.) were performed weekly after self-administration or when a compromised catheter was suspected. Animals not meeting criteria for patency (sedation within 5 s) were excluded.

## Cocaine self-administration under FR1 and PR schedules

After 5–7 days of surgical recovery, mice were trained to acquire cocaine self-administration in 3 hr sessions in operant conditioning chambers (Med. Associates, St Albans, VT). The position of the active and inactive nose ports was counterbalanced. Each response on the active operandum resulted in a cocaine infusion (i.e. FR1 schedule; 40–50 µl over 2.3–2.8, based on the weight of the mice) and illumination of the cue light above the active nose port for 5 s, followed by a 10 s timeout. Responses during timeouts were counted but had no consequence. Sessions ended early if 64 reinforcers were earned to avoid cocaine overdose. Each active nose poke delivered 0.5 mg/kg/infusion cocaine for 10 days until stable self-administration was acquired. Stable self-administration was defined as: ≥25 infusions and >2:1 active/inactive response ratio for 3 consecutive days, and less than 20% variability in daily drug infusions across two consecutive sessions. Mice that did not reach criteria following 10 days of training were excluded from further study. Following completion of the 10-day self-administration training, mice were injected with ESI-05 (3.3 or 10 mg/kg i.p.) or vehicle (0 mg/kg, i.p.) 10 min prior to self-administration testing under FR1 reinforcement in a Latin Square design. For i.p. injections, ESI-05 was first dissolved in DMSO to make 30 mg/ml stock solution, aliquoted and kept in a –80°C freezer. Before use, the stock solution was first mixed with same volume of Tween-80 and then diluted with saline (0.9% NaCl) to obtain a working solution (DMSO:Tween-80:saline = 1:1:18 solution). The same solution was used as vehicle control. Following drug treatment, mice self-administered cocaine as before until responding returned to stable levels, using the above criteria. Subsequent doses of ESI-05 or vehicle were then tested in a within-subjects design.

Additional groups of mice were trained under a PR schedule. After first establishing stable cocaine self-administration under an initial 10-day FR1 schedule, mice were switched to a PR schedule using 0.5 mg/kg cocaine. The PR schedule requires the subject to make an increasing number of operant responses (nose pokes) for each successive reward (cocaine infusion). The PR schedule used the series 1, 1, 2, 4, 5, 7, 10, 13, 16, 20, 25, 31, 38, 47, 57, 69, 84, 102, … following the equation: $N = 5 (e^{\text{infusion} \times 0.18} - 1)$ (*Richardson and Roberts, 1996*). The breakpoint is defined as the maximum number of lever presses completed for the last cocaine infusion prior to a 1 hr period during in which no infusions were obtained. When PR breakpoint stabilized for 3 consecutive days (defined as <15% variability in number of infusions over this period), mice received ESI-05 (3.3 or 10 mg/kg, i.p.) or vehicle (0 mg/kg, i.p.) 10 min before cocaine self-administration testing. Following drug treatment, mice self-administered cocaine as before until responding returned to stable levels, and subsequent doses of ESI-05 or vehicle were then tested in a within-subjects design.

To test the effects of genetic deletion of *Epac2* in dopamine neurons on cocaine self-administration, Epac2-cKO and WT mice were first trained to perform cocaine self-administration under an FR1 schedule as described above. After 10 days of FR1 cocaine self-administration, mice were switched to a PR reinforcement schedule (0.5 mg/kg/infusion). Active nose pokes, inactive nose pokes, and cocaine infusions under an FR1 schedule and cocaine infusions and breakpoint under a PR schedule were compared between WT and Epac2-cKO mice.

## Multiple-dose cocaine self-administration

Mice were first trained to self-administer cocaine under an FR1 schedule as described above. After criteria for stable cocaine-maintained responding were met, mice were switched to multiple-dose cocaine self-administration maintained by a full range of cocaine doses (see below) in a single dose-response session under FR1 reinforcement schedule. Training sessions under the multiple-dose schedule began with 30 min of the training dose cocaine (0.5 mg/kg/infusion, 3.53 s duration), then another six 30 min trials in which different doses of cocaine were presented in a descending order (*Anderson et al., 2018*) (1.0, 0.25, 0.125, 0.0625, 0.03125, and 0.0 mg/kg/infusion; the infusion durations were 7.07, 1.77, 0.88, 0.44, 0.22, 0 s, respectively), each dose trial was separated by a 1 min intertrial timeout period. ESI-05 pretreatment testing began once cocaine self-administration behavior stabilized, and stability was defined as: (1) less than 20% variation in total number of cocaine infusions for two consecutive test sessions; (2) the dose of cocaine that maintained maximal response rates varied by no more than one-half log unit over two consecutive test sessions (*Keck et al., 2014*; *Xi et al., 2017*). Mice received vehicle (0 mg/kg) or ESI-05 (3.3 or 10 mg/kg, i.p.) 10 min prior to the test session, and the test order for the various doses of ESI-05 or vehicle was counterbalanced. Following each vehicle or ESI-05 treatment, mice were allowed to self-administer under the multiple-dose

cocaine schedule until responding restabilized as defined above before subsequent doses of ESI-05 or vehicle were tested in a within-subjects design. Epac2-cKO and WT mice were also trained to self-administer cocaine using this protocol, and results were compared between groups.

## Oral sucrose self-administration

C57BL/6 mice were housed under a reverse light cycle without food restriction and handled for 3–6 days prior to procedures. Experiments began with daily 1 hr training sessions for 10 days, without any prior operant training. Each response on the active operandum results in sucrose delivery (10% sucrose; FR1 schedule; 35 µl over 1.98 s) and illumination of cue light (above nose port) for 5 s total and a 10 s timeout. Responses during timeout were counted but had no consequence. Sessions ended early if 64 reinforcers were earned. Stable sucrose self-administration was defined as: ≥30 infusions and >2:1 active/inactive response ratio for 4 consecutive days including the last day, and less than 20% variability in daily infusions across two consecutive sessions. Animals that did not reach criteria after 10 days of training were excluded from further study. Following completion of the 10-day self-administration training, mice were injected with ESI-05 (3.3 or 10 mg/kg, i.p.) or vehicle (0 mg/kg, i.p.) 10 min prior to self-administration testing under an FR1 schedule in a Latin Square design. Following drug treatment, animals self-administered sucrose as before until responding returned to stable levels, using the above criteria. Subsequent doses of ESI-05 or vehicle were then tested in a within-subjects design. Additional groups of mice were initially trained under FR1 reinforcement schedule. After stable FR1 reinforcement was established, animals were switched to PR reinforcement schedule. When PR breakpoint were stabilized for 3 consecutive days (defined as <15% variability in number of infusions over this period), mice received ESI-05 (3.3 or 10 mg/kg, i.p.) or vehicle injection 10 min before sucrose self-administration testing in a Latin Square design.

Epac2-cKO and WT mice were trained for sucrose self-administration under an FR1 schedule for 10 days. After FR1 sucrose self-administration training, mice were switched to PR schedule for 3 days. Active and inactive nose pokes, sucrose intake, and PR breakpoint were compared between groups.

## Locomotor activity test

To evaluate basal locomotion in WT and Epac2-cKO mice, a custom-made open-field (40 cm length ×40 cm width ×40 cm depth box) chamber was used. Mice were allowed to freely explore the arena during a 60 min test session. Locomotor activity was recorded using an automated video-tracking system (ANY-maze; Stoelting, Wood Dale, IL). Distance traveled in the chamber was recorded and calculated in 5 min intervals.

## In vivo fiber photometry to detect dopamine release in the NAc with GRAB$_{DA2m}$

A dual-wavelength fiber photometry system (Doric Lenses, Inc) was used to monitor fluorescence from dopamine-dependent GRAB$_{DA2m}$ and dopamine-independent mCherry reporters in the NAc of behaving mice (*Liu et al., 2022*). Briefly, a computer-controlled fiber photometry console and low-power LED drivers (LEDD_4) generated excitation light from two LEDs (465 and 560 nm). A lock-in amplifier sinusoidally modulated LED intensities at nonsynchronous frequencies (572.21 and 1017 Hz for 465 and 560 nm, respectively) to filter out potential fluorescence arising from overlapping excitation spectra of the reporters. A fluorescence MiniCube (Doric; FMC6_IE(400-410)_E(460-490)_F(500-550)_E(555-570)_F(580-680)_S) combined and directed the excitation lights through a low-autofluorescence fiber-optic patch cord (Doric; MFP_200/230/900–0.57_1.0_FCM_ZF1.25_LAF) and rotary joint (Doric; FRJ_200/230/LWMJ-0.57_1m_FCM_0.15_FCM). The patch cord was connected to an implanted fiber-optic cannula via a 1.25 mm mating sleeve (Doric; 1.25 mm with black cover; Sleeve_ZR_1.25-BK). Fluorescence emissions were returned through the same patch cord and MiniCube, separated, and directed to photoreceivers (Model 2151, DC low setting; Newport Corporation, Irvine, CA).

Before each experiment, the patch cord was photobleached with 465 and 560 nm excitation light to further reduce fiber-optic autofluorescence. The final power output at the tip of the patch cord was adjusted to 20 µW prior to recording. Data acquisition was controlled by Doric Neuroscience Studio software at a rate of 12.0 ksps, decimated by a factor of 100. The resulting signals were demodulated and low-pass filtered at 6 Hz. After attachment of the patch cord, mice were placed into an open-field chamber for at least 15 min to allow environmental habituation before recording commenced.

To reduce fluorescent reporter photobleaching, recording was performed using a 6 min 83% duty cycle (5 min on/1 min off). A baseline dopamine signal during open-field locomotion was recorded for 20 min. Then, mice received DCZ injections (100 µg/kg, subcutaneous) and were returned to the open-field chamber to record dopamine fluorescence for another 60 min.

The fiber photometry data were analyzed using custom code written in RStudio (GitHub - xiao-jieliu17/Fiber-photometry) (**Snarrenberg, 2022**). To calculate the ΔF/F time series, a linear fit was applied to the data and the fitted 560 nm signal was aligned and subtracted from the 465 nm signal, and then divided by the fitted 560 nm signal to yield ΔF/F values. For dopamine event frequency analysis, the processed ΔF/F values were transferred into ABF file using Minianalysis (Bluecell Co., Seoul, Korea). Then, the frequency of dopamine transients was analyzed using their Minianalysis software. The onset of an event was defined by a $Z$ score value larger than mean plus three times the standard deviation of the $Z$ scores of recorded fluorescence during the prior 0.5 s period, and the resultant peak frequency was compared across groups (**Meng et al., 2018**). After the completion of the experiments, the AAV expression in the VTA and NAc, and the localization of optic fiber implantation in the NAc was verified by immunohistochemistry.

## Acute brain slice preparation

WT and Epac2-cKO mice of either sex were used for FSCV, electrophysiological recordings, and scRT-PCR. Mice were anesthetized by isoflurane inhalation and decapitated. The brain was embedded in low-gelling-point agarose, and coronal striatal slices (200 µm thick) were cut using a vibrating slicer (Leica VT1200s, Nussloch, Germany), as described in our previous studies (**Vickstrom et al., 2020**). Slices were prepared in choline-based solution containing (in mM): 92 NMDG, 2.5 KCl, 1.25 $NaH_2PO_4$, 0.5 $CaCl_2$, 7 $MgSO_4$, 26 $NaHCO_3$, 25 glucose, 20 HEPES, 5 sodium ascorbate, 2 thiourea, and 3 sodium pyruvate. Slices were immediately transferred into the artificial cerebrospinal fluid (ACSF) (in mM): 119 NaCl, 2.5 KCl, 2.5 $CaCl_2$, 1 $MgCl_2$, 1.25 $NaH_2PO_4$, 26 $NaHCO_3$, and 10 glucose. Slices were allowed to recover at least 1 hr before recording. All solutions were saturated with 95% $O_2$ and 5% $CO_2$.

## Electrophysiology and scRT-PCR

Cell-attached and whole-cell recordings were performed to obtain the electrophysiological data and genetic expression data simultaneously. All equipment was cleaned with RNase Zap before each use. Glass pipettes were baked at 400°C for at least 4 hr to deactivate RNase. Glass pipettes (3–5 MΩ) were filled with RNase-free internal solution containing (in mM): 140 K-gluconate, 10 KCl, 10 HEPES, 0.2 EGTA, 2 $MgCl_2$, 4 Mg-ATP, 0.3 $Na_2GTP$, 10 $Na_2$-phosphocreatine (pH 7.2 with KOH). The internal solution was prepared with RNase-free water, aliquoted and stored at –80°C. Before each experiment, 10 µl recombinant RNase inhibitor (1 U/µl, Sigma-Aldrich, St Louis, MO) was added to 390 µl internal solution to increase RNA yield. Whole-cell and cell-attached patch-clamp recordings were made using patch-clamp amplifiers (Multiclamp 700B; Molecular Devices, San Jose, CA) under infrared differential interference contrast microscopy. Data acquisition and analysis were performed using DigiData 1440A and 1550B digitizers and the analysis software pClamp 10 (Molecular Devices). Signals were filtered at 2 kHz and sampled at 10 kHz. Action potential (AP) firing was recorded in the cell-attached configuration in the presence of CNQX (10 µM), D-AP5 (20 µM), and picrotoxin (50 µM) to block excitatory and inhibitory synaptic transmission, as described (**Vickstrom et al., 2021**). Junction potentials between the patch pipette and bath ACSF were nullified prior to obtaining a seal. Membrane capacitance was measured by Clampex software (Molecular Devices) using small amplitude hyperpolarizing and depolarizing steps (±5 mV). Series resistance (10–20 MΩ) was monitored throughout all recordings, and data were discarded if the resistance changed by more than 20%. All recordings were performed at 32 ± 1°C by using an automatic temperature controller (Warner Instruments LLC, Hamden, CT).

Cell contents were aspirated into the glass pipette by applying a gentle negative pressure at the end of each recording. The Gigaohm seal was maintained during the aspiration process to reduce external contamination. After aspiration, the pipette was slowly withdrawn from the cell. The cell contents were then dispensed into an RNase-free microcentrifuge tube with positive pressure. Complementary DNA (cDNA) was synthesized from the cytoplasmic mRNA using iScript Advanced cDNA Synthesis Kit (Bio-Rad Laboratories, Hercules, CA). Target genes were then amplified with two PCR steps using the following primer pairs. *Gapdh*, forward 5'- AGCTTGTCATCAACGGGAAG-3', reverse 5'-GTCATGAGCCCTTCCACAAT-3', 331 bp; *Th*, forward 5'-TGCCAGAGAAGGACAAGGTT

C-3', reverse 5'-CGATACGCCTGGTCAGAGA-3', 131 bp; *Gad2*, forward 5'-GTTCCTTTCCTGGTGAG TGC-3', reverse 5'-TGCATCAGTCCCTCCTCTCT-3', 266 bp; *Epac2*, forward 5'-TGGAACCAACTGG TATGCTG-3', reverse 5'-CCAATTCCCAGAGTGCAGAT-3', 102 bp; *Epac1*, forward 5'-GGACAAAGT CCCCTACGACA-3', reverse 5'- CTTGGTCCAGTGGTCCTCAT-3', 121 bp; *Slc17a6*, forward 5'-CGCTG CTTCTGGTTGTTGGC-3', reverse 5'-AAACCATCCCCGACAGCGTG-3', 186 bp. Each PCR amplification consisted of 35 cycles, 94°C for 30 s, 58°C for 45 s, and 72°C for 1 min. The products of the second PCR were analyzed in 1.5% agarose gels using ethidium bromide.

## Fast scan cyclic voltammetry

A glass-encased cylindrical carbon fiber (7 μm diameter, Goodfellow, Oakdale, PA) microelectrode with an exposed final length of 100–150 μm was lowered into four different sites of the striatum, DMS, DLS, NAc core, and NAc shell in striatal slices. The coordinates were determined by visual inspection of the slice with a mouse brain atlas (anteroposterior, 0.9–1.7 mm). The microelectrode was filled with solution containing 150 mM KCl. Triangular waveforms (holding at –0.4 V, from –0.4 to 1.3 V at 400 V/s) were applied every 100 ms using Demon software (*Yorgason et al., 2011*) and a Chem-Clamp Potentiostat (Dagan Corporation, Minneapolis, MN). A stimulating electrode was placed at ~100 μm from the carbon fiber microelectrode and dopamine release was evoked by a single electrical stimulus pulse (250 μA, 0.2 ms duration), or 5-pulse train stimuli at 100 Hz (*Liu et al., 2018b*). After the recordings, electrodes were calibrated in 0.1–2 μM dopamine in the ACSF. Dopamine release was evoked by a single pulse stimulation (250 μA, 0.2 ms duration) every 2 min, the effects of Epac2 or PKA agonists and/or antagonists on dopamine release were examined. After a stable baseline recording for 10 min, their effects on evoked dopamine release were examined for another 40 min. All recordings were performed at 32 ± 1°C by using an automatic temperature controller (Warner Instruments LLC).

## Immunofluorescence staining

Mice were anesthetized by ketamine (90 mg/kg, i.p.) and xylazine (10 mg/kg, i.p.) and transcardially perfused with 0.1 M sodium phosphate buffered saline (PBS) followed by 4% paraformaldehyde in 4% sucrose-PBS (pH 7.4). After perfusion, the brain was removed and post-fixed in the same fixative for 4 hr at 4°C, and was dehydrated in increasing concentrations of sucrose (20% and 30%) in 0.1 M PBS at 4°C. The fixed brain was frozen on dry ice, and coronal VTA or striatal sections (20 μm) were cut with a Leica cryostat (CM1860). VTA sections were incubated with primary antibody against tyrosine hydroxylase (TH, rabbit polyclonal, 1:300, Santa Cruz Biotechnology, Inc, Dallas, TX) at 4°C for 48 hr. After rinsing with PBS three times at 15 min each, VTA sections were then incubated in secondary antibody: Goat anti-rabbit IgG Alexa Fluor-488 for 4 hr at room temperature in the dark. Coronal striatal sections were rehydrated in PBS and permeabilized with 0.1% Triton X-100 in PBS followed by 640/660 deep-red fluorescent Nissl stain (N-21483) incubation for 20 min. Confocal imaging was performed using a Leica SP8 upright confocal microscope.

## Chemicals

ESI-05, 6-Bnz-cAMP sodium salt, H-89 dihydrochloride were purchased from Tocris Bioscience (Minneapolis, MN). Sp-8-BnT-cAMPS (S-220) was purchased from BioLog Life Science Institute (Farmingdale, NY). Water-soluble DCZ was purchased from Hello Bio Inc (Princeton, NJ). Cocaine HCl was provided by NIDA Drug Supply Program. All other common chemicals were purchased from Sigma-Aldrich.

## Statistical analysis

Data are presented as the mean ± SEM. Datasets were compared with either Student's *t*-test, paired *t*-test, one-way ANOVA, or two-way ANOVA on repeated measures followed by Tukey's post hoc analysis. For nonparametric PR breakpoint data, comparison was made with the Mann-Whitney rank test or Kruskal-Wallis one-way ANOVA on ranks followed by Dunn's post hoc analysis. Results were considered to be significant at $p < 0.05$.

## Acknowledgements

This work was supported by National Institutes of Health Grants F30MH115536 (to CRV), R01DA035217 (to QSL), R01DA047269 (to QSL), R01DA050180 (to DAB and QSL), and F31DA054759 (to VF). CRV and TJK are members of the Medical Scientist Training Program at MCW, which is partially supported

by a training grant from NIGMS T32-GM080202. We thank Dr Yulong Li (Peking University, Beijing, China) for gifting AAV9-hSyn-GRAB$_{DA2m}$ and the NIDA Drug Supply Program for providing the cocaine used in these studies.

## Additional information

### Funding

| Funder | Grant reference number | Author |
|---|---|---|
| National Institute on Drug Abuse | R01DA035217 | Qing-song Liu |
| National Institute on Drug Abuse | R01DA047269 | Qing-song Liu |
| National Institute of Mental Health | F30MH115536 | Casey R Vickstrom |
| National Institute on Drug Abuse | R01DA050180 | David A Baker |
| National Institute on Drug Abuse | F31DA054759 | Vladislav Friedman |
| National Institute of Mental Health | MH121454 | Qing-song Liu |

The funders had no role in study design, data collection and interpretation, or the decision to submit the work for publication.

### Author contributions

Xiaojie Liu, Conceptualization, Data curation, Formal analysis, Validation, Investigation, Visualization, Methodology, Writing - original draft, Project administration, Writing - review and editing; Casey R Vickstrom, Conceptualization, Data curation, Funding acquisition, Investigation, Methodology, Writing - original draft, Writing - review and editing; Hao Yu, Conceptualization, Data curation, Formal analysis, Investigation, Methodology, Writing - original draft, Writing - review and editing; Shuai Liu, Vladislav Friedman, Data curation, Formal analysis, Investigation, Writing - original draft, Writing - review and editing; Shana Terai Snarrenberg, Data curation, Software, Investigation, Methodology, Writing - review and editing; Lianwei Mu, Data curation, Formal analysis, Investigation, Methodology, Writing - review and editing; Bixuan Chen, Data curation, Investigation, Writing - review and editing; Thomas J Kelly, Data curation, Investigation, Methodology, Writing - review and editing; David A Baker, Conceptualization, Supervision, Funding acquisition, Writing - review and editing; Qing-song Liu, Conceptualization, Resources, Supervision, Funding acquisition, Investigation, Writing - original draft, Project administration, Writing - review and editing

### Author ORCIDs

Vladislav Friedman http://orcid.org/0000-0002-5567-2520
Qing-song Liu http://orcid.org/0000-0003-1858-1504

### Ethics

All animal maintenance and use were in accordance with protocols (AUA #2420) approved by the Institutional Animal Care and Use Committee of Medical College of Wisconsin. All surgery was performed under ketamine and xylazine anesthesia, and every effort was made to minimize suffering.

### Decision letter and Author response

Decision letter https://doi.org/10.7554/eLife.80747.sa1
Author response https://doi.org/10.7554/eLife.80747.sa2

## Additional files

### Supplementary files
• MDAR checklist

### Data availability

All data generated or analyzed during this study are included in the manuscript and supporting file; Source Data files have been provided for figures 1-7. Custom code used for the analysis of the fiber photometry data is available at https://github.com/xiaojieliu17/Fiber-photometry, (copy archived at swh:1:rev:ea959a0aed0ca49498c77d4d5479d004b6795545).

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
