## [Editor Report]

This manuscript reports that Epac2, a downstream effector of cAMP, positively regulates cocaine reward by altering dopamine release properties in the striatum. The study uses a broad range of technical approaches to thoroughly characterize the cellular and behavioral roles of Epac2 in mice. Together, these results provide convincing important insight into Epac2 as a potential presynaptic molecular target through which dopamine signaling and drug taking might be manipulated and is of interest to scientists studying dopamine transmission and substance use disorders.

---

## [Decision Letter]

**Decision letter after peer review:**

Thank you for submitting your article "Epac2 in midbrain dopamine neurons contributes to cocaine reinforcement via facilitation of dopamine release" for consideration by *eLife*. Your article has been reviewed by 3 peer reviewers, and the evaluation has been overseen by a Reviewing Editor and Kate Wassum as the Senior Editor. The following individual involved in the review of your submission has agreed to reveal their identity: Erin Calipari (Reviewer #1).

Essential revisions:

As outlined below, the reviewers were very positive about the rigor and potential significance of this manuscript and appreciated the incorporation of many different technical approaches to identify a role for presynaptic Epac2 in dopamine signaling and cocaine reinforcement/motivation. Although the reviewer critiques suggested potential experiments that could help to expand and clarify the mechanistic contributions of Epac2, the overall consensus was that these experiments are not essential for further consideration in *eLife*. However, at a minimum a revised manuscript should include:

1) Clarification of experimental questions and concerns raised by each reviewer, as well as the incorporation of suggested text changes.

2) Additional or more thorough discussion of limitations raised by the reviewers and implications of the work, including discussion of potentially non-specific effects of Epac2 manipulation in the NAc (Rev. 1), justification for the focus on the lateral NAc shell, and discussion of differences in Epac2 contributions to sucrose and cocaine reward (Rev. 2), and clarification of the working model presented in Figure 9 (Rev. 3).

*Reviewer #1 (Recommendations for the authors):*

1. I am confused about how on an FR1 schedule of reinforcement the infusions and lever presses are not the same. If they are different the animal is not actually on an FR1 schedule as every response does not actually result in an infusion of the drug. I would just change this to low effort schedule or something like that - no new experiments would have to be run.

2. While the experiments in Figure 6 are interesting, it is important to discuss what th`is means in the context of microcircuit work. It is possible that the effects of Epac2 could be having local effects selectively in dopamine terminals independent of somatic effects; however, the design is not able to distinguish between the effects on dopamine terminals and other neurotransmitter systems in the NAc such as GABA, acetylcholine, glutamate, and peptidergic release. If you knock out Epac2 in dopamine neurons, do you still get the effects of the drugs on local slices? The manuscript contains a lot of data, so it is not completely necessary to do these additional experiments, but this should be at the very least discussed in detail in the discussion.

*Reviewer #2 (Recommendations for the authors):*

Figures 1 and 3: ip injection of ESI-05 inhibit the global Epsac2. Given the extensive data based on dopamine neuron-specific manipulations, the data in these figures do not carry substantial weight. It might work better if some of the results are moved Suppl section.

Figures 3 and 4: It is interesting and somewhat surprising that Epac2 manipulations only influenced cocaine self-administration without affecting sucrose-motivated responses, given that dopamine is also needed for sucrose conditioning. Any insights warrant a discussion.

Figure 5: The lateral shell of the NAc was focused. Functionally, the medial shell is preferentially implicated in motivated behaviors. Some justification for focusing on the lateral shell will help.

Figures 7 and 8: The wild-type controls might need to be considered. Specifically, in addition to Epac2-cKO mice, will chemogenetic down/upregulation of dopamine release also influences mice without genetic modifications?

*Reviewer #3 (Recommendations for the authors):*

1. To address point #1 in the Public Review, a potential experiment for the authors to consider is to put back wild-type Epac2 protein to mutant VTA DA neurons (e.g. via viral gene overexpression) followed by measurement of DA release and cocaine SA. Authors may also consider testing if chemogenetic activation of DA neurons promotes sucrose SA, which may provide insights into the nature and significance of the rescue. If some or all of these experiments are not feasible or practical, detailed discussions and/or rationale need to be provided for clarification.

2. To address point #2 in the Public Review, authors may consider repeating the chemogenetic inhibition experiments in Epac2 cKO mice, and see if Epack2 deletion occluded or blunted effects of DA release inhibition on cocaine SA in mutant mice, or provide more detailed discussions or clarifications.

3. To address point #3 in the Public Review, authors should present the time course of DA transient changes following DCZ injection, and show more representative DA signal records.

4. Figure 9 working model may be revised by adding some intermediate steps between incoming action potential spikes and SV docking.

---

## [Author Response]

Reviewer #1 (Recommendations for the authors):1. I am confused about how on an FR1 schedule of reinforcement the infusions and lever presses are not the same. If they are different the animal is not actually on an FR1 schedule as every response does not actually result in an infusion of the drug. I would just change this to low effort schedule or something like that - no new experiments would have to be run.

The FR1 schedule has a brief timeout period (10 sec) following each active (rewarded) nose poke during which nose pokes do not result in cocaine infusions. As a result, the number of nose pokes is slightly higher than the number of cocaine infusions. We have clarified this in the legend of Fig. 1. However, to maintain convention, we did not change “FR1 schedule” to “low effort schedule”.

2. While the experiments in Figure 6 are interesting, it is important to discuss what this means in the context of microcircuit work. It is possible that the effects of Epac2 could be having local effects selectively in dopamine terminals independent of somatic effects; however, the design is not able to distinguish between the effects on dopamine terminals and other neurotransmitter systems in the NAc such as GABA, acetylcholine, glutamate, and peptidergic release. If you knock out Epac2 in dopamine neurons, do you still get the effects of the drugs on local slices? The manuscript contains a lot of data, so it is not completely necessary to do these additional experiments, but this should be at the very least discussed in detail in the discussion.

We agree with the reviewer that the Epac2 agonist acts locally on dopamine axonal terminals as the slice cutting removes the soma of dopamine neurons. The Cre-LoxP approach used here produced a complete knockout of Epac2 from entire midbrain dopamine neurons (soma and axonal terminals). Importantly, this dopamine neuron-specific knockout completely abolished the effect of the Epac2 agonist to increase dopamine release, suggesting that its effects are not mediated by other neurotransmitter systems. We have added the following discussion in the revised manuscript (Pages 16-17, lines 372-381) to further clarify and address this point.

“There is a possibility that the Epac2 agonist S-220 affected other neurotransmitter systems to indirectly modulate dopamine release. Optogenetic stimulation of cholinergic interneurons in the NAc induced dopamine release via activation of nicotinic acetylcholine receptors (nAChRs) (Cachope et al., 2012; Threlfell et al., 2012). GABA tonically depresses axonal dopamine release in the dorsal striatum via GABAA and GABAB receptors (Roberts et al., 2020). Thus, acetylcholine and GABA have indirect but major impacts on dopamine release. However, the S-220-induced enhancement of dopamine release was abolished by Epac2-cKO from dopamine neurons, suggesting that dopamine axonal terminals are the predominant site for the action of S-220. In future studies, it would be interesting to test whether the S-220-induced enhancement of dopamine release is altered by nAChR and GABA receptor blockers”.

Reviewer #2 (Recommendations for the authors):Figures 1 and 3: ip injection of ESI-05 inhibit the global Epsac2. Given the extensive data based on dopamine neuron-specific manipulations, the data in these figures do not carry substantial weight. It might work better if some of the results are moved Suppl section.

We have followed the reviewer’s advice and moved Figure 3 (ESI-05 on sucrose self-administration) into the supplemental section (now Figure 3—figure supplement 1). We would like to keep the original Figure 1 as we believe that ESI-05 inhibition of cocaine self-administration is a major finding of the manuscript.

Figures 3 and 4: It is interesting and somewhat surprising that Epac2 manipulations only influenced cocaine self-administration without affecting sucrose-motivated responses, given that dopamine is also needed for sucrose conditioning. Any insights warrant a discussion.

We have added the following discussion in the revised manuscript (Pages 15-16, lines 346-358):

“Cocaine and sucrose increase dopamine levels by reducing its uptake and increasing its release, respectively (Anderson and Pierce, 2005; Patriarchi et al., 2018). It was somewhat surprising that ESI-05 and Epac2 attenuated cocaine self-administration but did not significantly alter sucrose self-administration. However, similar differential modulation of cocaine vs. sucrose self-administration by pharmacological agents has been reported (Levy et al., 2007; Romieu et al., 2008; Mu et al., 2021). Although there is significant overlap in the neuronal circuits encoding food and drug rewards (Volkow et al., 2012), the majority of NAc neurons exhibited different firing patterns in response to food and cocaine rewards (Carelli, 2002). As mentioned in the Introduction, repeated cocaine exposure leads to decreased G_αi/o_ expression in the VTA and an up-regulation of cAMP signaling (Nestler et al., 1990; Striplin and Kalivas, 1992). It is likely that Epac2 is preferentially activated following chronic cocaine self-administration, which might explain why Epac2 disruption attenuated cocaine self-administration but not sucrose self-administration.”

Figure 5: The lateral shell of the NAc was focused. Functionally, the medial shell is preferentially implicated in motivated behaviors. Some justification for focusing on the lateral shell will help.

Our choice of the NAc lateral shell was driven by its denser innervation from VTA dopamine neurons compared to other NAc subregions as indicated by the mouse brain connectivity atlas from Allen Institute (https://connectivity.brain-map.org/) (PMID: 24695228) and our own preliminary studies.

Figures 7 and 8: The wild-type controls might need to be considered. Specifically, in addition to Epac2-cKO mice, will chemogenetic down/upregulation of dopamine release also influences mice without genetic modifications?

We have added the following discussion to address this question (Pages 18-19, lines 416-427) and related questions raised by Reviewer 3.

“We acknowledge the limitations of our DREADD experiments. First, Gs-DREADD stimulation was used to restore dopamine release in Epac2-cKO mice but did not restore Epac2 function. Further studies could determine whether viral expression of Epac2 in VTA dopamine neurons in Epac2-cKO mice could restore dopamine release and cocaine self-administration to the level of WT mice. Second, our DREADD studies were limited to experiments that were essential to test the central hypothesis that the Epac2-cKO-induced reduction of dopamine release contributes to the deficit in cocaine self-administration. Both mimicry and occlusion can provide insight into whether two biological processes occur by the same mechanism. We have shown that hM4D(Gi) inhibition of VTA dopamine neurons in WT mice mimicked the effects of Epac2-cKO on cocaine self-administration. A complementary experiment would be to determine whether hM4D(Gi) inhibition of dopamine neurons further decreases cocaine self-administration in Epac2-cKO mice or if the effect of hM4D(Gi) inhibition is occluded by Epac2-cKO.”

Reviewer #3 (Recommendations for the authors):1. To address point #1 in the Public Review, a potential experiment for the authors to consider is to put back wild-type Epac2 protein to mutant VTA DA neurons (e.g. via viral gene overexpression) followed by measurement of DA release and cocaine SA. Authors may also consider testing if chemogenetic activation of DA neurons promotes sucrose SA, which may provide insights into the nature and significance of the rescue. If some or all of these experiments are not feasible or practical, detailed discussions and/or rationale need to be provided for clarification.

Our lab has no direct experience with designing or producing viral vectors. It would take considerable time to collaborate with a Vector Core to design, pack and validate Epac2 overexpression in a cell-type specific manner and test its effects on cocaine self-administration in Epac2-cKO mice. Mouse Epac2 (Rapgef4) mRNA contains 3040 bp. Together with regulatory elements and EGFP, it may not be within the capacity of an AAV-mediated delivery system (~4.8 kb maximum), but it may be delivered with a lentiviral system. We are very interested in exploring this study at a future time. As mentioned above (question 1 of Public Review), testing whether chemogenetic activation of DA neurons *promotes* sucrose SA might not be feasible.

2. To address point #2 in the Public Review, authors may consider repeating the chemogenetic inhibition experiments in Epac2 cKO mice, and see if Epack2 deletion occluded or blunted effects of DA release inhibition on cocaine SA in mutant mice, or provide more detailed discussions or clarifications.

We have added discussion in the revised manuscript (pages 18-19, see also our response to the last question of Reviewer 2).

3. To address point #3 in the Public Review, authors should present the time course of DA transient changes following DCZ injection, and show more representative DA signal records.

We have presented the time course of DA transient changes following DCZ injection and showed more representative dopamine transients for fiber photometry recordings (Figure 6).

4. Figure 9 working model may be revised by adding some intermediate steps between incoming action potential spikes and SV docking.

We have revised the model by adding action potential firing (now Figure 8). However, we have removed SV docking as recommended above (question 4 of Public Review).